# ON STOCHASTIC CONTEXTUAL BANDITS WITH KNAPSACKS IN SMALL BUDGET REGIME

**Hengquan Guo and Xin Liu**[*]
School of Information Science and Technology
ShanghaiTech University
{guohq,liuxin7}@shanghaitech.edu.cn

## ABSTRACT

This paper studies stochastic contextual bandits with knapsack constraints (CBwK), where a learner observes a context, takes an action, receives a reward, and incurs a vector of costs at every round. The learner aims to maximize the cumulative rewards across $T$ rounds under the knapsack constraints with an initial budget of $B$. We study CBwK in the small budget regime where the budget $B = \Omega(\sqrt{T})$ and propose an Adaptive and Universal Primal–Dual algorithm (AUPD) that achieves strong regret performance: 1) AUPD achieves $\tilde{O}((1 + \frac{\nu^*}{\delta b})\sqrt{T})$ regret under the strict feasibility assumption without any prior information, matching the best-known bounds; 2) AUPD achieves $\tilde{O}(\sqrt{T} + \frac{\nu^*}{\sqrt{b}}T^{\frac{3}{4}})$ regret without strict feasibility assumption, which, to the best of our knowledge, is the first result in the literature. Here, the parameter $\nu^*$ represents the optimal average reward; $b = B/T$ is the average budget and $\delta b$ is the feasibility/safety margin. We establish these strong results through the adaptive budget-aware design, which effectively balances reward maximization and budget consumption. We provide a new perspective on analyzing budget consumption using the Lyapunov drift method, along with a refined analysis of its cumulative variance. Our theory is further supported by experiments conducted on a large-scale dataset.

## 1 INTRODUCTION

Stochastic contextual bandits with knapsacks (CBwK) is a general framework for online decision-making under resource-constrained applications and has been applied to a broad range of practical scenarios (e.g., resource-constrained recommendation Balakrishnan et al. (2018); Yang et al. (2020), clinic trials in healthcare Tewari & Murphy (2017); Tomkins et al. (2021), content moderation for healthy social platform Lykouris & Weng (2024); Lee et al. (2024), online advertisement platform Lucier et al. (2024); Feng et al. (2024); Gaitonde et al. (2022)). At each round $t$, the learner observes a context $x_t \in \mathcal{X}$ and chooses an action $a_t \in \mathcal{A}$. The learner then receives a reward $\mathcal{R}_t$ and also consumes $K$ types of resources or costs $\mathcal{C}_t$. The initial budget of the $k$th type of resource is $B_k$ and let $B = \min_k B_k$ and its average version be $b = B/T$. The interaction with the environment stops after $T$ rounds or when one of the resource budgets has been exhausted. The goal of the learner is to maximize the cumulative reward under the constrained resource budget.

CBwK is a contextual version of BwK, initially introduced in the literature Agrawal & Devanur (2014); Badanidiyuru et al. (2018), and has been extensively studied in Slivkins et al. (2023); Agrawal & Devanur (2016); Han et al. (2023); Chzhen et al. (2024). The work Slivkins et al. (2023) assumes online learning oracles are accessed to estimate the rewards and costs and leverages an inverse gap weighting method Abe & Long (1999); Foster & Rakhlin (2020) to balance exploration, reward acquisition, and cost consumption. However, this work assumes a large initial budget (or in a large budget regime) such that $B = \Omega(T)$. In many real-world applications, the budget is precious and scarce (i.e., $B \ll T$). For example, a store with a limited supply of $B$ items of a product provides customized offers to $T$ users in contextual dynamic pricing. Users arrive sequentially with different contexts, and the number of users is much larger than the number of items, i.e., $B \ll T$; In task

---

[*]Corresponding author

scheduling for the crowdsourcing platform, the tasks with different contexts arrive sequentially, and the platform decides if a task is good to assign to an expert or an ordinary worker. In this setting, the number of experts is much smaller than the number of tasks; In content moderation for healthy online review platforms, reviews are constantly posted on the platform, and the platform is required to filter out harmful reviews within a limited number of queries from human experts, where the query budget is much smaller than the number of reviews.

To solve problems with limited total budgets, the work Agrawal & Devanur (2016); Han et al. (2023) studied CBwK under either linear or general realizability assumptions on rewards/costs and attempt to relax the large budget requirement. However, these work still assume $B = \Omega(T^{\frac{3}{4}})$ due to the proposed two-stage strategies where an initial warm-up stage with $\sqrt{T}$ rounds is required in learning the optimal value for the subsequent decision-making. A very recent work Chzhen et al. (2024) further relaxed the budget requirement of $B = \Omega(\sqrt{T})$ by assuming the strict feasibility assumption (a.k.a the Slater's condition) and the prior knowledge of the feasibility/safety margin. However, obtaining such information can be quite (if not impossible) challenging in real-world scenarios. Besides, the work utilizes a double-tricking to learn the optimal stepsize in tracking the dual variable (the proxy for budget consumption) to achieve a good balance between rewards and resource consumption, which shares a similar flavor as the two-stage approach in Agrawal & Devanur (2016); Han et al. (2023). This raises a natural open question:

*Can we design an adaptive (ideally single-stage) algorithm to solve CBwK under the small budget $B = \Omega(\sqrt{T})$, achieving the universal and optimal performance regardless of the strict feasibility assumption and any prior knowledge about it?*

In this paper, we provide a positive answer to this question by introducing the Adaptive and Universal Primal–Dual algorithm (AUPD). Our contributions can be summarized as follows:

**Algorithm:** AUPD is motivated by the primal-dual approach which has also been widely explored in safe online learning Yu & Neely (2020); Slivkins et al. (2023); Guo & Liu (2024); Gangrade et al. (2024), but with an adaptive budget-aware design and a novel perspective from Lyapunov optimization. In the dual modular, we design the virtual queues $\{Q_t^{(k)}\}$, resemble scaled dual variables, to track the cumulative over-used budget consumption. Unlike previous studies in Agrawal & Devanur (2016); Han et al. (2023); Chzhen et al. (2024), we do not impose any upper bound or any conservative factors in the dual update. In the primal decision modular, we explicitly incorporate the knowledge of the initial budget $V = b\sqrt{T}$, in addition to the virtual queues that capture the over-used resource, to balance the reward and resource consumption. The scaled virtual queues $\{Q_t^{(k)}/V\}$ can be regarded as the estimators of the optimal dual variables, which avoids an explicit learning process used in Agrawal & Devanur (2016); Han et al. (2023); Chzhen et al. (2024). The budget-aware strategy and virtual queue design are the keys for AUPD to minimize regret and establish strong theoretical performance under the small budget regime without any prior information on the problem instance.

| Reference | Regret | Budget | Strict Feasibility |
|-----------|--------|--------|--------------------|
| SquareCBwK Han et al. (2023) | $\tilde{O}((1 + \frac{\nu^*}{\delta b})\sqrt{T})$ | $\Omega(T^{\frac{3}{4}})$ | *Required and known safety margin* |
| PGD Adaptive Chzhen et al. (2024) | $\tilde{O}((1 + \frac{\nu^*}{\delta b})\sqrt{T})$ | $\Omega(\sqrt{T})$ | *Required and known safety margin* |
| AUPD | $\tilde{O}((1 + \frac{\nu^*}{\delta b})\sqrt{T})$ | $\Omega(\sqrt{T})$ | *Required* |
| AUPD | $\tilde{O}(\sqrt{T} + \frac{\nu^*}{\sqrt{b}}T^{\frac{3}{4}})$ | $\Omega(\sqrt{T})$ | *Not Required* |

Table 1: Our results and most related works.

**Theoretical Results:** AUPD achieves the strong regret performance for CBwK in the small budget regime, regardless of the strict feasibility assumption (our results and most related work are summarized in Table 1): (i) When the strict feasibility assumption holds, AUPD achieves $\tilde{O}((1 + \frac{\nu^*}{\delta b})\sqrt{T})$ in the small budget regime $B = \Omega(\sqrt{T})$ regret under the general realizability assumption of reward and cost functions, where the parameter $\nu^*$ is the average performance of an optimal static policy and $\delta b$ is the feasibility/safety margin. This result matches the regret guarantee in Agrawal & Devanur (2016); Han et al. (2023); Chzhen et al. (2024). However, unlike in Agrawal & Devanur (2016); Han et al. (2023); Chzhen et al. (2024), Lyapunov drift analysis establishes an upper bound on the virtual queue (a proxy for the budget consumption process) without requiring any prior knowledge of the safety margin nor the extra implement parameter search steps. (ii) When the strict feasibility

assumption does not hold, AUPD achieves a worst-case regret of $\tilde{O}(\sqrt{T} + \frac{\nu^*}{\sqrt{b}}T^{\frac{3}{4}})$ in the small budget regime $B = \Omega(\sqrt{T})$. To the best of our knowledge, there are no existing results even in the large budget regime $B = \Omega(T)$ without the strict feasibility assumption.

**Experiments:** We evaluate AUPD using a real-world learning-to-rank dataset in the small budget regime. Our experimental results demonstrate that AUPD outperforms baseline algorithms and achieves the best performance under various initial budgets.

### RELATED WORK

Contextual bandits and bandits with knapsacks have been studied extensively. We only focus on the literature most related to ours.

**(Contextual) BwK: "large" budget regime and two-stage design.** Bandits with knapsacks (BwK) was initialized by Agrawal & Devanur (2014); Sankararaman & Slivkins (2018; 2021); Badanidiyuru et al. (2018), where the well-known optimistic exploration with the primal-dual design has been extensively studied in Agrawal & Devanur (2014; 2019); Agrawal et al. (2016); Badanidiyuru et al. (2018); Immorlica et al. (2022) to reduce BwK to unconstrained bandits. BwK has been further extended to the contextual version of BwK in Agrawal & Devanur (2016); Badanidiyuru et al. (2014); Agrawal et al. (2016); Li & Stoltz (2022); Slivkins et al. (2023); Han et al. (2023) under different realizability assumptions for rewards and costs, such as linear class, logistic function class or even general function class. When the linear structure is imposed on the rewards and costs, the work Agrawal & Devanur (2016) integrates the primal-dual design with LinUCB exploration and designs a two-stage learning method to achieve the regret of $\tilde{O}((1 + \frac{\nu^*}{b})\sqrt{T})$ when $B = \Omega(T^{\frac{3}{4}})$. Inspired by the estimation-to-decision framework for contextual bandits in Foster & Rakhlin (2020), CBwK under general realizability conditions is studied in Slivkins et al. (2023); Han et al. (2023) by assuming access to online learning oracles. However, the work Slivkins et al. (2023) requires a large budget assumption where $B = \Omega(T)$ and the work Han et al. (2023) still assumes $B = \Omega(T^{\frac{3}{4}})$ as it also utilizes the two-stage method to search the optimal value $Z = \nu^*/b$ as in Agrawal & Devanur (2016), both of these works assume prior knowledge about the strict feasibility margin.

**Contextual BwK: "small" budget regime and strict feasibility assumption.** There exist two recent work Kim et al. (2023); Chzhen et al. (2024) that attempt to break the barrier of $B = \Omega(T^{\frac{3}{4}})$, where they achieve the regret of $\tilde{O}((1 + \frac{\nu^*}{b})\sqrt{T})$ under the relaxed budget $B = \Omega(\sqrt{T})$. However, these works rely on specific assumptions: Kim et al. (2023) requires a null action (which is a more stringent requirement), while Chzhen et al. (2024) assumes strict feasibility. Besides, Kim et al. (2023) assumes linear rewards and costs, and Chzhen et al. (2024) presumes the safety margin is known, allowing the learner to make conservative decisions. Although Chzhen et al. (2024) claims that their proposed algorithm is a direct primal-dual approach (unlike the previous two-stage designs), they still need to implement a doubling trick to search for the optimal dual variables, which shares a similar spirit to the two-stage algorithm in Agrawal & Devanur (2016).

**(Contextual) BwK: "flexible" budget regime and without hard-stopping.** There exists another line of literature where the interaction can continue even after the budget is exhausted Kumar & Kleinberg (2022); Bernasconi et al. (2024a;b), which we call a "flexible" budget regime. Kumar & Kleinberg (2022) consider non-monotonic or replenishable resource utilization for (non-contextual) stochastic bandits with knapsacks, where a "null action" is introduced to allow the budgets to be replenished. Bernasconi et al. (2024b) extended bandits with replenishable knapsack constraints into adversarial scenarios and achieved "best-of-both-worlds" regret guarantees. Very recently, Bernasconi et al. (2024a) studied stochastic and adversarial bandits with general constraints, where constraint violations are allowed and no hard stopping is imposed. They proposed weakly adaptive primal and dual algorithms that achieve tight regret bounds in both stochastic and adversarial settings. However, extending these results to the hard stop setting might require additional and dedicated procedures.

## 2 STOCHASTIC CONTEXTUAL BANDITS WITH KNAPSACKS

In this section, we introduce Stochastic Contextual Bandits with Knapsacks (CBwK) defined by $\{\mathcal{X}, \mathcal{A}, r, \boldsymbol{c}, \boldsymbol{B}\}$, where $\mathcal{X}$ is the context set (a countable set), $\mathcal{A}$ is the action set (a finite set), $r : \mathcal{X} \times \mathcal{A} \to [0, R]$ is the reward function, $\boldsymbol{c} : \mathcal{X} \times \mathcal{A} \to [0, C]^K$ are the cost functions, and

$\boldsymbol{B} = [B_1, \ldots, B_K]$ is the initial total budget vector. Without loss of generality[1], we assume the uniform budget $B_k = B$, $\forall k \in [K]$. At the beginning of every round $t \in [T]$, the learner observes a context $x_t$ that is randomly generated from the context set $\mathcal{X}$ according to an unknown probability distribution $p_x$. The learner takes an action $a_t \in \mathcal{A}$ according to a policy defined by $\pi : \mathcal{X} \to \mathcal{A}$, and then observes the noisy reward, i.e., $\mathcal{R}_t(x_t, a_t)$, and the noisy costs, denoted by $\mathcal{C}_t(x_t, a_t)$. Given the initial budgets $\boldsymbol{B}$, the interaction terminates once any type of resource is exhausted or it reaches the end of time horizon $T$. Note that the context distribution, reward, and cost functions are all unknown to the learner. We present the following assumption on them.

**Assumption 1** *The context $\{c_t\}$ are i.i.d. across rounds. There exists reward function $r : \mathcal{X} \times \mathcal{A} \to [0, R]$ and cost functions $\mathbf{c} : \mathcal{X} \times \mathcal{A} \to [0, C]^K$ satisfy that $r(x, a) = \mathbb{E}[\mathcal{R}_t(x_t, a)|x_t = x]$ and $\boldsymbol{c}(x, a) = \mathbb{E}[\mathcal{C}_t(x_t, a)|x_t = x]$, $\forall x \in \mathcal{X}, \ a \in \mathcal{A}$.*

Let $a_t^{\pi_t}$ be the action taken by the policy $\pi_t(x_t)$ given context $x_t$ in round $t$. For the given instance $\{x_t\}_t$ drawn from a certain distribution, the learner's objective is to design dynamic policies that maximize the cumulative rewards over horizon $T$ under *the knapsack constraints*:

$$\max \ \sum_{t=1}^{T} r(x_t, a_t^{\pi_t}) \ \ \text{s.t.} \ \sum_{t=1}^{T} \boldsymbol{c}(x_t, a_t^{\pi_t}) \leqslant \boldsymbol{B}. \tag{1}$$

Note it could be quite challenging to solve (1) because the reward, cost, and context distributions are all unknown to the learner. The budget constraint further complicates the problem because all actions are coupled over the time horizon, making it difficult to balance the reward acquisition and budget consumption. To evaluate the performance of a policy, we introduce the definition of regret.

**Regret:** We begin with an offline problem by assuming the full knowledge of rewards, costs, and context distribution. Define $\boldsymbol{b} := \boldsymbol{B}/T = [b, \ldots, b]$ with $b = B/T$, the offline problem in (1) can be reformulated as follows

$$\max_{\pi} \ \sum_{x \in \mathcal{X}, a \in \mathcal{A}} p_x r(x, a) \pi(x, a) \tag{2}$$

$$\text{s.t.} \ \sum_{x \in \mathcal{X}, a \in \mathcal{A}} p_x c^{(k)}(x, a) \pi(x, a) \leqslant b, \ \forall k \in [K] \tag{3}$$

$$\sum_{a \in \mathcal{A}} \pi(x, a) = 1, \ \pi(x, a) \geqslant 0, \forall x \in \mathcal{X}, \tag{4}$$

where $\pi(x, a)$ can be viewed as the probability of taking action $a$ on context $x$, and $p_x$ is the probability that context $x$ is sampled in each round. The next lemma shows that the optimal value of (2)–(4) serves as an upper bound on that of (1). The detailed proof can be found in Appendix A.

**Lemma 1** *Under Assumption 1, let $\nu^*$ be the optimal value of the offline problem (2)–(4) and $OPT$ be the expected reward of the optimal dynamic policies to the problem (1), respectively. We have $OPT \leqslant T\nu^*$.*

Now we define (pseudo)-regret based on the baseline above for an algorithm as follows

$$\text{Regret}(T) = T\nu^* - \mathbb{E}\left[\sum_{t=1}^{T} r(x_t, a_t)\right]. \tag{5}$$

The expectation is taken w.r.t. randomness from $\{a_t\}_t$ drawn by the algorithm and the environment.

Before presenting our algorithm and analyzing the regret performance, we first introduce the modeling and estimation of reward and cost functions.

**Learning Oracles:** We assume there exist online learning oracles for reward and cost functions such that the corresponding estimators are either optimistic or pessimistic and the cumulative estimation errors can be bounded. Specifically, given the historical feedback information $\{\mathcal{R}_s(x_s, a_s), \mathcal{C}_s(x_s, a_s)\}_{s=1}^{t-1}$ at every round $t$, the online learning oracles will output the estimators of reward and costs $\hat{r}_t : \mathcal{X} \times \mathcal{A} \to [0, R]$ and $\check{c}_t : \mathcal{X} \times \mathcal{A} \to [0, C]^K$ that satisfy the following assumptions.

---

[1]If the initial budgets $\{B_k\}$ are different, we can always make them identical with the scaling factor $B_k / \min_k B_k$ for every type of resource.

**Assumption 2** *There exist online learning oracles* $\{\mathcal{O}\}_{r,\boldsymbol{c}}$ *such that the reward and cost estimators* $\hat{r}_t(x,a)$ *and* $\check{c}_t^{(k)}(x,a)$ *satisfy the following conditions with a high probability at least* $1-p$:

$$\mathcal{E} = \left\{ \begin{array}{l} 0 \leqslant \hat{r}_t(x,a) - r(x,a) \leqslant 2\varepsilon_t(x,a,p), \\ 0 \leqslant c^{(k)}(x,a) - \check{c}_t^{(k)}(x,a) \leqslant 2\varepsilon_t(x,a,p), \ \forall k \in [K], \ x \in \mathcal{X}, \ a \in \mathcal{A}, \ t \in [T] \end{array} \right\},$$

*where* $p = 1/T^2$, *and* $U(T,p) := \sum_{t=1}^{T} \varepsilon_t(x_t,a_t,p) = O(\sqrt{T}\log(T/p))$.

This assumption describes the performance of learning oracles, a common condition that can be met in contextual bandits Abbasi-yadkori et al. (2011); Filippi et al. (2010); Foster et al. (2018); Han et al. (2023); Chzhen et al. (2024). When the reward and cost functions belong to the linear class, the classical online least-square regression oracle satisfy Assumption 2 and $\varepsilon_t(x,a,p)$ is simply the upper/lower confidence bound Abbasi-yadkori et al. (2011). When the reward and cost functions belong to the generalized linear class, the online maximum-likelihood estimate oracles satisfy Assumption 2 and $\varepsilon_t(x,a,p)$ is the generalized upper/lower confidence bound Filippi et al. (2010). When the reward and cost functions are general and do not have a good structure, the weighted online regression estimators still satisfy Assumption 2 and $\varepsilon_t(x,a,p)$ can be calculated efficiently via binary search method Foster et al. (2018).

# 3 ADAPTIVE AND UNIVERSAL PRIMAL–DUAL ALGORITHM FOR CBWK

In this section, we introduce AUPD, an *adaptive*, single-stage algorithm for solving CBwK, designed to achieve the *universal* regret guarantee under both the absence and presence of the Slater's condition in the small budget regime. AUPD does not require any prior knowledge except the initial budgets **B** and the time horizon $T$. The design of AUPD is motivated by the primal-dual optimization approach Wright (1997); Bertsimas & Tsitsiklis (1997), incorporating an adaptive budget-aware design with a new perspective from Lyapunov optimization. We utilize the virtual queues to estimate the over-used budget consumption and carefully choose the tradeoff parameter to balance the rewards and budget consumption. AUPD algorithm comprises the following key components:

- **Budget-Aware Decision-Making:** At each round $t$, AUPD first obtains estimated reward and cost functions from the learning oracles $\{\mathcal{O}\}_{r,\boldsymbol{c}}$, which are used to construct optimistic/pessimistic estimators $\hat{r}_t$ and $\check{c}_t$. The action is chosen to maximize $\hat{r}_t - \sum_k Q_t^{(k)}\check{c}_t^{(k)}/V$, inspired by "Reward - Lyapunov Drift" in Lyapunov optimization, where the Lyapunov drift $\sum_k[(Q_{t+1}^{(k)})^2 - (Q_t^{(k)})^2]$, is approximated by $\sum_k Q_t^{(k)}\check{c}_t^{(k)}/V$ according to the virtual queues update in (7). This resembles the primal-dual method, where the primal modular is to maximize the approximated Lagrangian function
$$L(x_t,a) := r(x_t,a) - \sum_k \lambda^{(k)}(c^{(k)}(x_t,a) - b)),$$
  where the Lagrange multiplier $\lambda^{(k)}$ is approximated by $Q_t^{(k)}/V$, and the reward and cost functions are approximated by the optimistic/pessimistic estimators. The key design is the budget-aware trade-off parameter $V = b\sqrt{T}$ (resembling an adaptive learning rate). It is different from the traditional Lyapunov optimization method where the trade-off parameter is only related to the time horizon $T$. Intuitively, when the budget is small, $V$ is small, and it prompts more conservative decisions; otherwise, it prompts relatively optimistic decisions.
  Unlike the approaches in Agrawal & Devanur (2016); Han et al. (2023); Chzhen et al. (2024), which require the extra parameter search stage, AUPD enjoys a single-stage and direct greedy structure in decision-making. Besides, the primal decision module implicitly learns the context distribution, as the decision in (6) tends to avoid overspending the resources on contexts with low rewards and high costs.

- **Oracles Update and Budget Pacing:** After observing the noisy reward $\mathcal{R}_t(x_t,a_t)$ and noisy costs $\mathcal{C}_t(x_t,a_t)$, AUPD feeds them into learning oracles (e.g., online weighted regression oracles) to construct estimators for future rounds. The other key design in AUPD is budget pacing, where we design the virtual queues to track the cumulative over-used budget consumption $\sum_{s=1}^{t}(\check{c}_t^{(k)}(x_t,a_t) - b)$. The concept of virtual queues originates from queueing theory and is widely used in networking and operations research in Hajek (1982); Neely (2010); Eryilmaz & Srikant (2012). In a real queueing

system, customers arrive, receive service, and leave, with the queues capturing the carryover effect and representing the number of waiting customers. The real queues motivate the design of virtual queues $Q_t^{(k)}$ in our setting, it represents the cumulative overuse of a resource, where the "arrival" corresponds to the current resource consumption and the "service" corresponds to the average budget. When the virtual queues increase, it implies that we might have spent too many resources so far, and it encourages a relatively conservative decision. The virtual queues can be regarded as the scaled dual variables in the primal-dual design in Agrawal & Devanur (2016); Han et al. (2023); Chzhen et al. (2024).

However, unlike these works, which either impose upper bounds or conservative factors in the dual update that require a two-stage learning process, strict feasibility assumption, or the knowledge of feasibility/safety margin, the virtual queues update in AUPD is again natural and direct. This is the key reason we do not require strict feasibility assumptions or any prior information about the feasibility/safety margin. Furthermore, by treating the virtual queues as Markovian processes, we can use Lyapunov drift analysis to establish their upper bound, which can then be translated into a strong lower bound on the stopping time.

---

**Algorithm 1 Adaptive and Universal Primal–Dual Algorithm for CBwK**

---

1: **Initialization:** $\lambda = 1$, $Q_1^{(k)} = 0$, $\forall k \in [K]$ and $V = b\sqrt{T}$, learning oracles set $\{\mathcal{O}\}_{r,\mathbf{c}}$.
2: **for** $t = 1, \cdots, T$, **do**
3:     **Parameters Estimation:** Given a context $x_t$, estimate the reward functions $\hat{r}_t(x_t, a)$ and the cost function $\check{c}_t(x_t, a)$ from learning oracles $\{\mathcal{O}\}_{r,\mathbf{c}}$.
4:     **Budget-Aware Decision-Making:** Take the action $a_t$ such that

$$a_t = \underset{a \in \mathcal{A}}{\operatorname{argmax}} \ V\hat{r}_t(x_t, a) - \sum_k Q_t^{(k)} \check{c}_t^{(k)}(x_t, a). \tag{6}$$

5:     **Feedback and Oracles Update:** Observe noisy reward $\mathcal{R}_t(x_t, a_t)$ and noisy costs $\mathcal{C}_t(x_t, a_t)$ and feed them to $\{\mathcal{O}\}_{r,\mathbf{c}}$.
6:     **Budget Pacing:** Update virtual queues as follows

$$Q_{t+1}^{(k)} = \left[Q_t^{(k)} + \check{c}_t^{(k)}(x_t, a_t) - b\right]^+, \ \forall k \in [K]. \tag{7}$$

7: **end for**

---

In summary, AUPD provides a novel algorithm design template and theoretical analysis for CBwK. The budget-aware decision-making and virtual queue-based budget-pacing are crucial in developing a fully adaptive algorithm to achieve a strong regret performance in the small budget regime.

## 4 MAIN RESULTS

In this section, we analyze the regret performance for AUPD in Algorithm 1. We begin by stating the strict feasibility assumption for the performance analysis.

**Assumption 3** *There exists a constant $\delta \in (0, 1]$ such that a feasible solution $\pi$ to the optimization problem (2)–(4) satisfies $\sum_{x \in \mathcal{X}, a \in \mathcal{A}} p_x c^{(k)}(x, a)\pi(x, a) \leqslant b(1 - \delta)$, $\forall k \in [K]$.*

The term $\delta b$ plays a similar role with the Slater's constant in optimization. However, the term $\delta b$ differs from the traditional Slater's constant because the definition $b = B/T$ implies it is both budget and time horizon related. Note that the existence of null action is a special case of this assumption with $\delta = 1$. This assumption has been used in the literature of BwK Agrawal & Devanur (2014; 2016); Badanidiyuru et al. (2014; 2018); Chzhen et al. (2024).

Now, we are ready to present the theoretical results of our algorithm, which are given in an order-wise sense. The detailed proof and parameters are in the appendix.

**Theorem 1** *Under Assumptions 1 and 2, AUPD achieves the following regret in the small budget regime $B = \Omega(\sqrt{T})$ that*

$$\text{Regret}(T) = \tilde{O}\left(\sqrt{T} + \frac{\nu^*}{\sqrt{b}}T^{\frac{3}{4}}\right).$$

When the additional strict feasibility assumption in Assumption 3 holds, AUPD achieves the following regret in the small budget regime

$$\text{Regret}(T) = \tilde{O}\left((1 + \frac{\nu^*}{\delta b})\sqrt{T}\right).$$

**Remark 1** *When the strict feasibility assumption holds, AUPD achieves $\tilde{O}((1 + \frac{\nu^*}{\delta b})\sqrt{T}) = \tilde{O}\left(\sqrt{T} + \frac{1}{\delta}T\nu^*T^{\frac{1}{2}-\alpha}\right)$ regret in the small budget regime $B = T^\alpha = \Omega(\sqrt{T})$ under the general realizability assumption of reward and cost functions. The parameter $\nu^*/\delta b$ in the regret captures the effect of knapsack constraints, where it could be large when the safety margin $\delta b$ is small, indicating the challenge of distinguishing the budget consumption among context-action pairs. The typical and practical setting would have $T\nu^* = \Theta(T^\alpha)$ (i.e., $T\nu^* = \Theta(B)$) because it represents "one unit of reward earned by consuming one unit of cost". In this setting, we can get the classical and optimal $\tilde{O}(\sqrt{T})$ regret. This result matches the regret guarantee in Han et al. (2023); Chzhen et al. (2024). Unlike these works, AUPD does not require the doubling trick or other parameter search stages, thus providing a more direct and adaptive structure.*

*When the strict feasibility assumption does not necessary to hold, AUPD achieves a worst-case theoretical guarantee of $\tilde{O}(\sqrt{T} + \frac{\nu^*}{\sqrt{b}}T^{\frac{3}{4}}) = \tilde{O}\left(\sqrt{T} + T\nu^*T^{\frac{1}{4}-\frac{\alpha}{2}}\right)$ in the small budget regime, which, to be the best of our knowledge, is the first result for CBwK without strict feasibility assumption even when $B = \Omega(T)$. Intuitively, the additional term $\frac{\nu^*}{\sqrt{b}}T^{\frac{3}{4}}$ is due to the "hard" problem instance. When $T\nu^* = \Theta(T^\alpha) = \Theta(B)$, i.e., one unit of reward for each unit of cost, AUPD achieves the regret $\tilde{O}(T^{\frac{3}{4}})$ performance, instead of the classical regret $\tilde{O}(\sqrt{T})$. This might imply that the CBwK instance, without a strict feasibility assumption, is indeed challenging.*

*Finally, it is worth emphasizing that all these results are achieved with a single AUPD algorithm, without any tailored adjustment or any prior knowledge of the problem instance. This demonstrates that AUPD is quite adaptive and universal for CBwK.*

**Remark 2** *The classical lower bound for CBwK is $\Omega(\sqrt{T})$ in Agrawal & Devanur (2016) derived by reducing the constrained contextual bandits into unconstrained ones. However, this lower bound did not capture the effect of knapsack constraints. To our knowledge, the most relevant lower bound for CBwK is from Chzhen et al. (2024). With the assumption of strict feasibility, Section E in Chzhen et al. (2024) provides a problem-dependent lower bound of $\Omega((1 + \nu^*/b)\sqrt{T})$ for CBwK with $B = \Omega(\sqrt{T})$. Therefore, our regret bound is tight when the assumption of strict feasibility holds. However, no existing lower bounds are reported without the assumption of strict feasibility, which is an interesting direction for future work.*

To establish these strong results, we need to carefully analyze the budget consumption processes. The key is to identify when the interaction terminates/stops and how much cumulative rewards AUPD gains during the process. When the strict feasibility assumption holds, we provide a new perspective on the stopping time (i.e., the first time when any type of resource budget is exhausted) and a refined analysis of the cumulative variance of budget consumption. Our analysis treats the virtual queue update as a Markovian process and leverages the Lyapunov-drift analysis to establish the expected upper bound on the virtual queues (i.e., the over-consumed budgets), which is translated to be the lower bound of the stopping time. Without the strict feasibility assumption, we directly establish the upper bound of the virtual queues. These bounds will be used to establish the regret bounds in Theorem 1, as we detailed in the next section.

## 5 THEORETICAL ANALYSIS

In this section, we provide a detailed proof of Theorem 1. To state these results, we first decompose the regret defined in (5) according to the stopping time $\tau$. The stopping time is defined as the first time when one of the resource budgets is exhausted

$$\tau = \underset{\tau' \in [T]}{\operatorname{argmin}} \left\{ \tau' \mid \exists k, \ \sum_{t=1}^{\tau'} c^{(k)}(x_t, a_t) \geqslant B \right\}. \tag{8}$$

### 5.1 REGRET DECOMPOSITION

Let $\pi^*$ be the optimal solution to the baseline problem (2)–(4) and $a_t^*$ be the optimal actions sampling from it, i.e., $a_t^* \sim \pi^*$. We decompose the regret as follows

$$\text{Regret}(T) \leqslant \mathbb{E}\left[\sum_{t=1}^T r(x_t, a_t^*) - \sum_{t=1}^T r(x_t, a_t)\right] \tag{9}$$

$$\leqslant \underbrace{\nu^* \mathbb{E}[T - \tau]}_{\text{Regret after stopping}} + \underbrace{\mathbb{E}\left[\sum_{t=1}^\tau r(x_t, a_t^*) - r(x_t, a_t)\right]}_{\text{Regret before stopping: Regret}(\tau)}$$

The decomposition includes two parts: "regret after stopping" and "regret before stopping". The former one is simply bounded by the remaining rounds $\times$ the optimal value $\nu^*$; the latter one denoted by $\mathbb{E}[\text{Regret}(\tau)]$ is the difference between our policy and the optimal one.

**Regret via Lyapunov Drift Analysis:** As discussed, we provide a new perspective on analyzing the regret via Lyapunov drift analysis, which can be used to bound both "regret before stopping" and "regret after stopping". We view $\{Q_t\}$ as a stochastic/Markovian process and study its connection with regret. Let $L_t = \|\boldsymbol{Q}_t\|_2^2/2 = \sum_k (Q_t^{(k)})^2/2$ be the Lyapunov function and $\Delta_t = L_{t+1} - L_t$ be its drift. Further let $\mathbb{E}_{\mathcal{H}_t}[\cdot] = \mathbb{E}[\cdot|\mathcal{H}_t]$, where $\mathcal{H}_t = [x_t, \hat{r}_t, \check{c}_t, \boldsymbol{Q}_t]$ We establish the following key lemma that bridges the one-step regret and Lyapunov drift. The detailed proof is in Appendix B.1.

**Lemma 2** *Under Assumptions 1 and 2, AUPD in Algorithm 1 establishes that for any feasible policy $\pi$ to (2)–(4) with $a \sim \pi$ that*

$$\mathbb{E}_{\mathcal{H}_t}\left[\text{Regret}(x_t, a) + \frac{\Delta_t}{V} - \frac{1}{V}\sum_k Q^{(k)}(c^{(k)}(x_t, a) - b)\right]$$

$$\leqslant \mathbb{E}_{\mathcal{H}_t}\left[\frac{2}{T^2} + 2\varepsilon_t(x_t, a_t, p) + \frac{1}{2V}\sum_k \left(\check{c}_t^{(k)}(x_t, a_t) - b\right)^2 + \sum_k \frac{2Q^{(k)}(1+b)}{V}\frac{\mathbb{P}\left(\mathcal{E}^c\right)}{\mathbb{P}\left(\mathcal{E}\right)}\right],$$

*where $\text{Regret}(x_t, a) = r(x_t, a) - r(x_t, a_t)$ and the event $\mathcal{E}$ is defined in Assumption 2.*

This Lemma is the key lemma that establishes the bound of "one-step regret + Lyapunov drift". It bridges the analysis to bound both "regret before/after stopping" as we will demonstrate in the following. Note that the lemma holds without strict feasibility assumption in Assumption 3.

### 5.2 REGRET BEFORE STOPPING

Letting $a = a_t^* \sim \pi^*$ in Lemma 2 and ignoring the low probability event $\mathcal{E}^c$, then the inequality suggests that "one-step regret + Lyapunov drift" is upper bounded by three related terms: the optimal budget consumption $\sum_k (c^{(k)}(x_t, a_t^*) - b)$, the single-step estimation error $\varepsilon_t(x_t, a_t, p)$, the estimated consumption resource $\sum_k (\check{c}_t^{(k)}(x_t, a_t) - b)^2$. Bounding "regret before stopping" $\mathbb{E}[\text{Regret}(\tau)]$ requires establishing their cumulative counterparts: the expected optimal budget consumption is always negative according to the definition; the cumulative estimation error is bounded by Assumption 2; the estimated consumption resource is the most important part and we provide a refined analysis that bound this term by $O((Tb + Tb^2)/V)$, which is one of the key components in proving the strong results in Theorem 1. The result of "regret before stopping" is summarized in the following lemma and the detailed proof is in Appendix B.2.

**Lemma 3** *Under Assumptions 1 and 2, the budget-aware optimistic exploration algorithm in Algorithm 1 achieves*

$$\mathbb{E}\left[\text{Regret}(\tau)\right] = O\left(\sqrt{T}\log(T) + \frac{K(Tb + Tb^2)}{V}\right).$$

### 5.3 REGRET AFTER STOPPING

In the regret decomposition, the "regret after stopping" is bounded by $\nu^* \mathbb{E}[T - \tau]$. To minimize this regret, it is crucial to establish a "large" lower bound on the stopping time $\tau$, ideally depleting the

budget only when it is very close to $T$. To (lower) bound the stopping time $\tau$, we need to analyze the behavior of the virtual queue because it captures the over-consumed budget against the average usage $b$ for the round $t$.

Recall we have the virtual queue update $Q_{t+1}^{(k)} = \max(Q_t^{(k)} + \check{c}_t^{(k)}(x_t, a_t) - b, 0)$. Let $M_\tau^{(k)} = \sum_{t=1}^{\tau}(c^{(k)}(x_t, a_t) - \check{c}_t^{(k)}(c_t, a_t))$, then for certain budget $k'$ that is first to be exhausted, we have

$$Q_{\tau+1}^{(k')} + b\tau + M_\tau^{(k')} \geqslant \sum_{t=1}^{\tau} c^{(k')}(x_t, a_t). \tag{10}$$

From learning oracle errors in Assumption 2, we already have a high probability upper bound for $M_\tau^{(k)}, \forall k \in [K]$, then we can consequently define a virtual stopping time such that $\tau_0 = \operatorname{argmin}_{\tau' \in [T]}\{\tau' \mid Q_{\tau'+1}^{(k')} + b\tau' + \tilde{O}(\sqrt{\tau}) \geqslant B\}$, where $k'$ denotes the resource that was first depleted. The virtual stopping time is the first time when the upper bound of budget consumption in (10) is greater than $B$. The inequality (10) indicates that the value $Q_{t+1}^{(k')}$ reflects how soon the algorithm will deplete the resource. Combined with the definition of the stopping time in (8), we immediately have that the true stopping time is lower bounded by the virtual stopping time, i.e., $\tau \geqslant \tau_0$. Now we only need to establish the lower bound on $\tau_0$, and the key is to prove an upper bound for the virtual queue length at the stopping time, which can be provided through Lyapunov drift analysis.

**Lyapunov drift analysis for establishing $\tau_0$:** As discussed, we view $\{Q_t\}$ as a stochastic/Markovian process and study its convergence or upper bound via Lyapunov analysis. When the strict feasibility assumption in Assumption 3 holds, from Lemma 2, we establish a "negative drift" of the Lyapunov drift function, implying a "small" expected upper bound of the virtual queues in the following lemma. Without Assumption 3, we establish a slightly worse upper bound.

**Lemma 4** *Under Assumptions 1 and 2, when $B = \Omega(\sqrt{T})$, AUPD in Algorithm 1 achieves that*

$$\mathbb{E}\left[\sum_k Q_t^{(k)}\right] = O(\sqrt{KVT}), \ \forall t \in [T],$$

*with the additional Assumption 3, AUPD guarantees that*

$$\mathbb{E}\left[\sum_k Q_t^{(k)}\right] = O(\sqrt{K}V/\delta b), \ \forall t \in [T]. \tag{11}$$

The above lemma demonstrates that the virtual queues are "stable" and relatively "small", indicating that our algorithm utilizes the resources effectively and does not terminate early. Intuitively, the underlying reasons behind virtual queues staying within a "small" region because, upon detecting over-consumption (the virtual queues increases), the algorithm would make conservative decisions to reduce the queue length. Now, we are ready to establish the expected upper bound on the remaining rounds in the following lemma, which is the key to establishing the regret performance.

**Lemma 5** *Under Assumptions 1 and 2, the expected remaining round in the small budget regime $B = \Omega(\sqrt{T})$ under AUPD in Algorithm 1 satisfies that*

$$\mathbb{E}[T - \tau] = \tilde{O}\left(\sqrt{KVT}/b + \sqrt{T}/b\right),$$

*with the additional Assumption 3, the expected remaining round under AUPD satisfies that*

$$\mathbb{E}[T - \tau] = \tilde{O}\left(\sqrt{K}V/\delta b^2 + \sqrt{T}/b\right).$$

**Proving Theorem 1:** Lemma 3 demonstrates that the algorithm's performance approaches the optimal policy before stopping, while Lemma 5 ensures that the algorithm would not exhaust resources and terminate prematurely. Combine them into (9), we immediately have the regret bound for two cases. Specifically, for the worst-case without any feasibility assumptions, we have

$$\begin{aligned}
\text{Regret}(T) &\leqslant \nu^* \mathbb{E}[T - \tau] + \mathbb{E}[\text{Regret}(\tau)] \\
&= \tilde{O}\left(\sqrt{KVT}\nu^*/b + \sqrt{T}\nu^*/b + \sqrt{T} + K(Tb + Tb^2)/V\right).
\end{aligned}$$

With the strict feasibility assumption in Assumption 3, we have

$$\text{Regret}(T) \leqslant \nu^* \mathbb{E}[T - \tau] + \mathbb{E}[\text{Regret}(\tau)]$$
$$= \tilde{O}\left(\sqrt{K}V\nu^*/\delta b^2 + \sqrt{T}\nu^*/b + \sqrt{T} + K(Tb + Tb^2)/V\right).$$

Let $V = b\sqrt{T}$ and we prove Theorem 1.

## 6 EXPERIMENTS

In this section, we validate our algorithm through numerical experiments using the large-scale learning-to-rank dataset Qin & Liu (2013) in the small budget regime. We compare our algorithm AUPD with SquareCBwK Han et al. (2023), an oracle–based primal–dual algorithm under $B = \Omega(T^{\frac{3}{4}})$, and PGD Adaptive Chzhen et al. (2024), a primal–dual algorithm with doubling trick for learning the optimal stepsize that claims to achieve optimal performance under $B = \Omega(\sqrt{T})$. The large-scale learning-to-rank dataset MSLR-WEB30k Qin & Liu (2013) contains $31,278$ queries. Each query includes various document-query contexts, each with a dimensionality of 136, and there are 20 documents/arms. The reward function $r(x, a)$ is defined as the relevance of each document-context pair as collected in the dataset. We construct the expected cost of each arm $c(x, a)$ by uniformly drawing from the interval $[0, 5]$, and these values remain fixed throughout each trial. The observations are corrupted with Gaussian noise $\mathcal{N}(0, 0.05)$. All the algorithms utilize gradient-boosted tree regression as the learning oracle for the reward function and the empirical mean for the cost function. We set the time horizon $T = 5000$ and vary the budgets $B = \{100, 600, 1000\}$ to represent the budget regime $\{\Theta(\sqrt{T}), \Theta(T^{\frac{3}{4}}), \Theta(T)\}$, respectively. The interaction terminates once the budget is exhausted, incurring zero reward and cost for the remaining rounds. The further details on the experiments and hyperparameters can be found in Appendix D. Our experimental results are shown in Figure 1. These results are obtained by averaging over 50 trials and are reported with a 95% confidence interval.

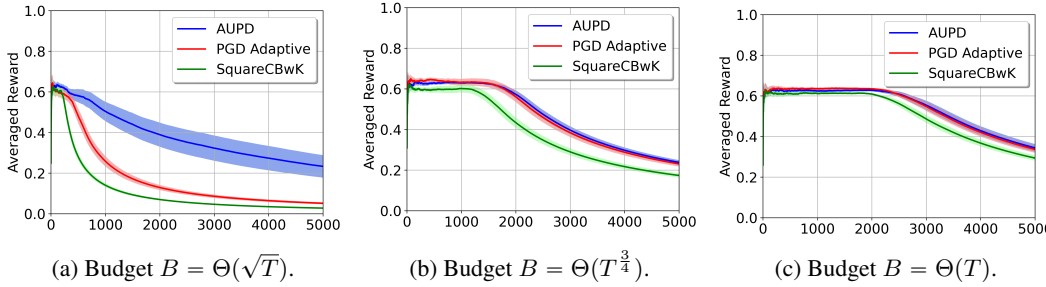

(a) Budget $B = \Theta(\sqrt{T})$.  (b) Budget $B = \Theta(T^{\frac{3}{4}})$.  (c) Budget $B = \Theta(T)$.

Figure 1: Experiments under different budget regimes.

From Figure 1, we observe our algorithm AUPD achieves the best performance compared to SquareCBwK and PGD Adaptive under various budget regimes. Especially when the budget is relatively small, as shown in Figures 1a, AUPD outperforms the baselines by a large margin. This justifies that the joint design of budget-aware decision-making and virtual queue-based budget pacing is very adaptive and effective to balance the reward and budget consumption.

## 7 CONCLUSIONS

In this paper, we studied stochastic contextual bandits with knapsack constraints under the small budget regime. We introduced an Adaptive and Universal Primal–Dual Algorithm and provided a new perspective of Lyapunov drift analysis to establish strong theoretical guarantees under the small budget regime $B = \Omega(\sqrt{T})$: AUPD achieves the best well-known regret under the strict feasibility assumption; AUPD achieves the first regret result without the strict feasibility assumption. The experiments further confirmed our theoretical results.

ACKNOWLEDGEMENTS

The work was partly supported by the National Nature Science Foundation of China under grant 62302305 and Shanghai Sailing Program 22YF1428500.

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

## A    PROOF OF LEMMA 1

The proof is similar to that in Devanur et al. (2011). Recall the problem in (1) and let $\hat{\pi}_t^*$ be the adaptive optimal solution:

$$\max \; \sum_{t=1}^{T} r(x_t, a_t^{\pi_t})$$

$$\text{subject to:} \; \sum_{t=1}^{T} \boldsymbol{c}(x_t, a_t^{\pi_t}) \leqslant \boldsymbol{B}$$

Recall the context $x_t$ is i.i.d. across rounds and $r(x, a)$ and $\boldsymbol{c}(x, a)$ are the expected functions given $x$. We have

$$\mathbb{E}\left[\sum_{t=1}^{T}\sum_{a} r(x_t, a)\pi_t(x_t, a)\right] = \sum_{t=1}^{T}\mathbb{E}\left[\mathbb{E}\left[\sum_{a} r(x_t, a)\pi_t(x_t, a)|\mathcal{H}_{t-1}\right]\right]$$

$$= \sum_{t=1}^{T}\mathbb{E}\left[\sum_{x}\sum_{a} p_x r(x, a)\mathbb{E}\left[\pi_t(x, a)|\, x_t = x, \mathcal{H}_{t-1}\right]\right]$$

$$= \mathbb{E}\left[\sum_{x}\sum_{a} p_x r(x, a)\sum_{t=1}^{T}\mathbb{E}\left[\pi_t(x, a)|\, x_t = x, \mathcal{H}_{t-1}\right]\right]$$

$$= \sum_{x}\sum_{a} p_x r(x, a)\sum_{t=1}^{T}\mathbb{E}\left[\pi_t(x, a)|\, x_t = x\right]$$

Similarly, we have

$$\mathbb{E}\left[\sum_{t=1}^{T}\sum_{a} c^{(k)}(x_t, a)\pi_t(x_t, a)\right] = \sum_{x}\sum_{a} p_x c^{(k)}(x, a)\sum_{t=1}^{T}\mathbb{E}\left[\pi_t(x, a)|\, x_t = x\right], \; \forall k \in [K].$$

Define $\hat{\pi}^*(x, a) = \frac{1}{T}\sum_{t=1}^{T}\mathbb{E}\left[\hat{\pi}_t^*(x, a)|\, x_t = x\right]$, where $\hat{\pi}_T^*$ is a feasible solution to (2)–(4) because $\hat{\pi}^*$ is a feasible solution to (1). Therefore, we have

$$\mathbb{E}\left[\sum_{t=1}^{T}\sum_{a} r(x_t, a)\hat{\pi}_t^*(x_t, a)\right] = T\sum_{x}\sum_{a} p_x r(x, a)\hat{\pi}^*(x, a)$$

$$\leqslant T\nu^*.$$

## B    REGRET ANALYSIS

For the sake of simplicity, we initially let the reward and cost upper bounds $R = C = 1$ in Assumption 1 before delving into detailed proofs.

### B.1    PROOF OF LEMMA 2

In Algorithm 1, the action $a_t$ is the optimal solution such that for any $a \in \mathcal{A}$,

$$\hat{r}_t(x_t, a) - \frac{1}{V}\sum_{k} Q_t^{(k)}\check{c}_t^{(k)}(x_t, a) \leqslant \hat{r}_t(x_t, a_t) - \frac{1}{V}\sum_{k} Q_t^{(k)}\check{c}_t^{(k)}(x_t, a_t).$$

Add $\text{Regret}(x_t, a) = r(x_t, a) - r(x_t, a_t)$ on both sides and rearrange these terms, we have

$$\text{Regret}(x_t, a) \tag{12}$$

$$\leqslant (r(x_t, a) - \hat{r}_t(x_t, a)) + (\hat{r}_t(x_t, a_t) - r(x_t, a_t)) + \frac{1}{V}\sum_{k} Q_t^{(k)}\left(\check{c}_t^{(k)}(x_t, a) - \check{c}_t^{(k)}(x_t, a_t)\right).$$

According to the virtual queue update

$$Q_{t+1}^{(k)} = \max\left(Q_t^{(k)} + \check{c}_t^{(k)}(x_t, a_t) - b, 0\right), \ \forall k.$$

The following inequality holds for the Lyapunov drift that

$$\Delta_t := \frac{1}{2}\sum_k \left(Q_{t+1}^{(k)}\right)^2 - \frac{1}{2}\sum_k \left(Q_t^{(k)}\right)^2 \leqslant \frac{1}{2}\sum_k (2Q_t^{(k)} + \check{c}_t^{(k)}(x_t, a_t) - b)(\check{c}_t^{(k)}(x_t, a_t) - b)$$

$$= \sum_k Q_t^{(k)}\left(\check{c}_t^{(k)}(x_t, a_t) - b\right) + \sum_k \frac{1}{2}\left(\check{c}_t^{(k)}(x_t, a_t) - b\right)^2.$$

Combine this inequality with (12), we can establish the following key bound on the "Regret + Lyapunov drift":

$$\text{Regret}(x_t, a) + \frac{\Delta_t}{V} \leqslant (r(x_t, a) - \hat{r}_t(x_t, a)) + (\hat{r}_t(x_t, a_t) - r(x_t, a_t))$$

$$+ \frac{1}{V}\sum_k Q_t^{(k)}\left(\check{c}_t^{(k)}(x_t, a) - b\right) + \frac{1}{2V}\sum_k \left(\check{c}_t^{(k)}(x_t, a_t) - b\right)^2. \tag{13}$$

To proceed, we recall the following event defined in Assumption 2, which holds with a probability of at least $1 - p$:

$$\mathcal{E} = \left\{\begin{matrix} 0 \leqslant \hat{r}_t(x, a) - r(x, a) \leqslant 2\varepsilon_t(x, a, p), \\ 0 \leqslant c^{(k)}(x, a) - \check{c}_t^{(k)}(x, a) \leqslant 2\varepsilon_t(x, a, p), \ \forall k \in [K], \ x \in \mathcal{X}, \ a \in \mathcal{A}, \ t \in [T] \end{matrix}\right\}.$$

Recall $p = 1/T^2$, now taking the conditional expectation $\mathbb{E}[\ \cdot\ |\mathcal{H}_t = h]$ on (13), where $h = [x, \hat{f}, \check{c}, Q]$, we obtain that

$$\mathbb{E}\left[\text{Regret}(x_t, a)|\mathcal{H}_t = h\right] + \mathbb{E}\left[\frac{\Delta_t}{V}|\mathcal{H}_t = h\right]$$

$$\leqslant \mathbb{E}\left[(r(x_t, a) - \hat{r}_t(x_t, a)) + (\hat{r}_t(x_t, a_t) - r(x_t, a_t))|\mathcal{H}_t = h, \mathcal{E}^c\right] + 2\varepsilon_t(x_t, a_t, p)$$

$$+ \mathbb{E}\left[\frac{1}{V}\sum_k Q_t^{(k)}\left(\check{c}_t^{(k)}(x_t, a) - b\right) + \frac{1}{2V}\sum_k \left(\check{c}_t^{(k)}(x_t, a_t) - b\right)^2|\mathcal{H}_t = h\right]$$

$$\leqslant \frac{2}{T^2} + 2\varepsilon_t(x_t, a_t, p) + \mathbb{E}\left[\frac{1}{2V}\sum_k \left(\check{c}_t^{(k)}(x_t, a_t) - b\right)^2|\mathcal{H}_t = h\right]$$

$$+ \mathbb{E}\left[\frac{1}{V}\sum_k Q_t^{(k)}\left(\check{c}_t^{(k)}(x_t, a) - b\right)|\mathcal{H}_t = h\right]$$

$$\leqslant \frac{2}{T^2} + 2\varepsilon_t(x_t, a_t, p) + \mathbb{E}\left[\frac{1}{2V}\sum_k \left(\check{c}_t^{(k)}(x_t, a_t) - b\right)^2|\mathcal{H}_t = h\right]$$

$$+ \frac{1}{V}\sum_k Q^{(k)}\mathbb{E}\left[(c^{(k)}(x_t, a) - b)\mid \mathcal{H}_t = h, \mathcal{E}\right] + \sum_k \frac{Q^{(k)}(1 + b)}{V}\mathbb{P}\left(\mathcal{E}^c\right)$$

$$\leqslant \frac{2}{T^2} + 2\varepsilon_t(x_t, a_t, p) + \mathbb{E}\left[\frac{1}{2V}\sum_k \left(\check{c}_t^{(k)}(x_t, a_t) - b\right)^2|\mathcal{H}_t = h\right]$$

$$+ \frac{1}{V}\sum_k Q^{(k)}\mathbb{E}\left[(c^{(k)}(x_t, a) - b)|\mathcal{H}_t = h\right] + \sum_k \frac{2Q^{(k)}(1 + b)}{V}\frac{\mathbb{P}\left(\mathcal{E}^c\right)}{\mathbb{P}\left(\mathcal{E}\right)}, \tag{14}$$

where the second and third inequalities hold because of the definition of event $\mathcal{E}$. The last inequality holds because the context is independent of $\mathcal{H}_t$ and the distribution of $x_t$ does not change conditioned on $\mathcal{H}_t$. Therefore, we have

$$\mathbb{P}\left(x_t = x|\mathcal{H}_t = h, \mathcal{E}\right) = \mathbb{P}\left(x_t = x|\mathcal{E}\right).$$

Then we calculate that

$$
\begin{aligned}
\mathbb{P}\left(x_t = x | \mathcal{E}\right) - \mathbb{P}\left(x_t = x\right) &= \frac{\mathbb{P}(x_t = x, \mathcal{E}) - \mathbb{P}\left(x_t = x\right)\mathbb{P}\left(\mathcal{E}\right)}{\mathbb{P}(\mathcal{E})} \\
&= \frac{\mathbb{P}\left(x_t = x\right)\left(\mathbb{P}(\mathcal{E}|x_t = x) - \mathbb{P}\left(\mathcal{E}\right)\right)}{\mathbb{P}(\mathcal{E})} \\
&\leqslant \mathbb{P}\left(x_t = x\right)\frac{1 - \mathbb{P}\left(\mathcal{E}\right)}{\mathbb{P}\left(\mathcal{E}\right)},
\end{aligned}
$$

which implies that

$$
\sum_k Q^{(k)}\mathbb{E}\left[(c^{(k)}(x_t, a) - b)|\mathcal{H}_t = h, \mathcal{E}\right] - \sum_k Q^{(k)}\mathbb{E}\left[(c^{(k)}(x_t, a) - b)|\mathcal{H}_t = h\right]
$$

$$
\leqslant \sum_k Q^{(k)}(1 + b)\frac{1 - \mathbb{P}\left(\mathcal{E}\right)}{\mathbb{P}\left(\mathcal{E}\right)} = \sum_k Q^{(k)}(1 + b)\frac{\mathbb{P}\left(\mathcal{E}^c\right)}{\mathbb{P}\left(\mathcal{E}\right)}.
$$

## B.2 PROOF OF LEMMA 3

Since the action $a$ could be any action in $\mathcal{A}$, we can let $a = a_t^* \sim \pi^*$ in (14) to obtain:

$$
\mathbb{E}\left[\text{Regret}(x_t, a_t^*)|\mathcal{H}_t = h\right] + \mathbb{E}\left[\frac{\Delta_t}{V}|\mathcal{H}_t = h\right]
$$

$$
\leqslant \frac{2}{T^2} + 2\varepsilon_t(x_t, a_t, p) + \mathbb{E}\left[\frac{1}{2V}\sum_k \left(\check{c}_t^{(k)}(x_t, a_t) - b\right)^2 |\mathcal{H}_t = h\right] + \sum_k \frac{2Q^{(k)}(1 + b)}{V}\frac{\mathbb{P}\left(\mathcal{E}^c\right)}{\mathbb{P}\left(\mathcal{E}\right)}
$$

$$
\tag{15}
$$

where the inequality holds because the action $a_t^*$ satisfies the constraint in (3) such that $\mathbb{E}\left[(c^{(k)}(x_t, a_t^*) - b)|\mathcal{H}_t = h\right] \leqslant 0$. We further take the expectation on both sides of the inequality and then take summation from $t = 1$ to $\tau$ that

$$
\mathbb{E}\left[\sum_{t=1}^{\tau}\text{Regret}(x_t, a_t^*)\right] + \mathbb{E}\left[\sum_k \frac{(Q_{\tau+1}^{(k)})^2}{2V} - \sum_k \frac{(Q_1^{(k)})^2}{2V}\right]
$$

$$
\leqslant \frac{2\tau}{T^2} + 2\sum_{t=1}^{\tau}\varepsilon_t(x_t, a_t, p) + \mathbb{E}\left[\sum_{t=1}^{\tau}\frac{1}{2V}\sum_k \left(\check{c}_t^{(k)}(x_t, a_t) - b\right)^2\right] + \mathbb{E}\left[\sum_{t=1}^{\tau}\sum_k \frac{2Q^{(k)}(1 + b)}{V}\frac{\mathbb{P}\left(\mathcal{E}^c\right)}{\mathbb{P}\left(\mathcal{E}\right)}\right].
$$

Since $Q_1^{(k)} = 0$, $\forall k \in [K]$, $\mathbb{P}(\mathcal{E}^c) \leqslant 1/T^2$ and $Q_t^{(k)} \leqslant Q_{t-1}^{(k)} + 1 \leqslant T$, $\forall k \in [K]$, $t \in [T]$, we conclude that

$$
\mathbb{E}\left[\sum_{t=1}^{\tau}\text{Regret}(x_t, a_t^*)\right] \leqslant \frac{2}{T} + 2\sum_{t=1}^{T}\varepsilon_t(x_t, a_t, p) + 4K(1 + b) + \mathbb{E}\left[\sum_{t=1}^{\tau}\frac{1}{2V}\sum_k \left(\check{c}_t^{(k)}(x_t, a_t) - b\right)^2\right]
$$

$$
\leqslant \frac{2}{T} + 2U(T, p) + 4K(1 + b) + \mathbb{E}\left[\sum_{t=1}^{\tau}\frac{1}{2V}\sum_k \left(\check{c}_t^{(k)}(x_t, a_t) - b\right)^2\right],
$$

where the first inequality holds since stopping time $\tau \leqslant T$, and the second inequality comes from the learning oracle assumption. Finally, we complete the proof by providing a *simple but refined analysis* on the cumulative budget consumption that

$$
\mathbb{E}\left[\sum_{t=1}^{\tau}\frac{1}{2V}\sum_k \left(\check{c}_t^{(k)}(x_t, a_t) - b\right)^2\right] \leqslant \mathbb{E}\left[\sum_{t=1}^{\tau}\sum_k \frac{\check{c}_t^{(k)}(x_t, a_t)^2 + b^2}{V}\right]
$$

$$
\leqslant \frac{1}{V}\mathbb{E}\left[\sum_{t=1}^{\tau}\sum_k \check{c}_t^{(k)}(x_t, a_t)\right] + \frac{\tau K b^2}{V},
$$

where the second inequality holds because $\check{c}_t(x_t, a_t)$ is bounded by 1. Moreover, we have

$$
\mathbb{E}\left[\sum_{t=1}^{\tau}\check{c}_t^{(k)}(x_t, a_t)\right] = \mathbb{E}\left[\sum_{t=1}^{\tau}\check{c}_t^{(k)}(x_t, a_t) - c^{(k)}(x_t, a_t)\right] + \mathbb{E}\left[\sum_{t=1}^{\tau}c^{(k)}(x_t, a_t)\right] \leqslant 1 + B_k = 1 + Tb,
$$

where the last inequality holds because of Assumption 2 and the definition of stopping time $\tau$. Therefore, we have

$$\mathbb{E}\left[\sum_{t=1}^{\tau}\frac{1}{2V}\sum_k\left(\check{c}_t^{(k)}(x_t,a_t)-b\right)^2\right]\leqslant\frac{K(1+Tb+\tau b^2)}{V}\leqslant\frac{K(1+Tb+Tb^2)}{V}.$$

By combining all these facts, we derive an upper bound for the regret incurred before stopping.

$$\mathbb{E}\left[\text{Regret}(\tau)\right]\leqslant\frac{2}{T}+2U(T,p)+4K(1+b)+\frac{K(1+Tb+Tb^2)}{V}$$

$$\leqslant 2+2C_0\sqrt{T}\log(T/p)+4K(1+b)+K\left(\frac{1}{b\sqrt{T}}+1+b\right)\sqrt{T}$$

$$\leqslant(6K+4Kb)+2C_0\sqrt{T}\log(T/p)+K(2+b)\sqrt{T}$$

$$=C_1+2C_0\sqrt{T}\log(T/p)+K(2+b)\sqrt{T},$$

where $C_1=6K+4Kb$ and the second inequality holds due to the value of $V$, while the third inequality is a consequence of the fact that $K\geqslant 1$.

### B.3 PROOF OF LEMMA 4

#### B.3.1 LYAPUNOV DRIFT ANALYSIS

From (14), we have established the Lyapunov drift

$$\mathbb{E}\left[\sum_k\frac{(Q_{t+1}^{(k)})^2}{2}-\sum_k\frac{(Q_t^{(k)})^2}{2}|\mathcal{H}_t\right]$$

$$\leqslant\frac{2V}{T^2}+2V\varepsilon_t(x_t,a_t,p)+\mathbb{E}\left[-V\text{Regret}(x_t,a)+\frac{1}{2}\sum_k\left(\check{c}_t^{(k)}(x_t,a_t)-b\right)^2\mid\mathcal{H}_t\right]$$

$$+\sum_k Q^{(k)}\mathbb{E}\left[(c^{(k)}(x_t,a)-b)|\mathcal{H}_t\right]+\sum_k\frac{2Q^{(k)}(1+b)}{V}\frac{\mathbb{P}\left(\mathcal{E}^c\right)}{\mathbb{P}\left(\mathcal{E}\right)}$$

$$\leqslant 4V+\mathbb{E}\left[-V\text{Regret}(x_t,a)+\frac{1}{2}\sum_k\left(\check{c}_t^{(k)}(x_t,a_t)-b\right)^2\mid\mathcal{H}_t\right]$$

$$+\sum_k Q^{(k)}\mathbb{E}\left[(c^{(k)}(x_t,a)-b)|\mathcal{H}_t\right]+\sum_k\frac{2Q^{(k)}(1+b)}{V}\frac{\mathbb{P}\left(\mathcal{E}^c\right)}{\mathbb{P}\left(\mathcal{E}\right)}\tag{16}$$

$$\leqslant 6V+\mathbb{E}\left[\frac{1}{2}\sum_k\left(\check{c}_t^{(k)}(x_t,a_t)-b\right)^2\mid\mathcal{H}_t\right]-\sum_k Q^{(k)}\delta b+\sum_k\frac{2Q^{(k)}(1+b)}{V}\frac{\mathbb{P}\left(\mathcal{E}^c\right)}{\mathbb{P}\left(\mathcal{E}\right)}$$

$$\leqslant 6V+\mathbb{E}\left[\frac{1}{2}\sum_k\left(\check{c}_t^{(k)}(x_t,a_t)-b\right)^2\mid\mathcal{H}_t\right]-\sum_k Q^{(k)}\delta b+\sum_k\frac{4Q^{(k)}(1+b)}{VT^2}$$

$$\leqslant 6V+\mathbb{E}\left[\frac{1}{2}\sum_k\left(\check{c}_t^{(k)}(x_t,a_t)-b\right)^2\mid\mathcal{H}_t\right]-\sum_k Q^{(k)}\delta b+\frac{4K(1+b)}{VT}\tag{17}$$

where the second inequality holds because $\varepsilon_t(x_t,a_t,p)\leqslant 1$ as specified by the estimators' and the function's upper bounds; the third inequality holds because of the upper bounds of reward function and cost estimators and the "Slater condition" in Assumption 3 that there exists a feasible policy such that

$$\mathbb{E}\left[c^{(k)}(x_t,a)-b\mid\mathcal{H}_t=h\right]\leqslant-\delta b,\ \forall k\in[K].$$

### B.4 CBwK WITHOUT STRICT FEASIBILITY ASSUMPTIONS

For CBwK without strict feasibility assumptions, we can set $a=a^*\sim\pi^*$ in (16). This allows us to obtain an inequality similar to (17), replacing $-\sum_k Q^{(k)}\delta b$ with 0, we can then directly take the

expectation on both sides to obtain:

$$\mathbb{E}\left[\sum_k \frac{(Q_{t+1}^{(k)})^2}{2} - \sum_k \frac{(Q_t^{(k)})^2}{2}\right] \leqslant 6V + \mathbb{E}\left[\frac{1}{2}\sum_k \left(\check{c}_t^{(k)}(x_t, a_t) - b\right)^2\right] + \frac{4K(1+b)}{VT}$$

$$\leqslant 6V + \mathbb{E}\left[\frac{1}{2}\sum_k \left(\check{c}_t^{(k)}(x_t, a_t) + b^2\right)\right] + \frac{4K(1+b)}{VT},$$

where the inequality holds since $Q^{(k)}$, $\delta$, $b \geqslant 0$, $(a-b)^2 \leqslant a^2 + b^2$ and the value of $\check{c}_t^{(k)}(x, a)$ are bounded. Sum this inequality from 1 to $\tau$, then we have

$$\mathbb{E}\left[\sum_k \frac{(Q_{\tau+1}^{(k)})^2}{2} - \sum_k \frac{(Q_1^{(k)})^2}{2}\right] \leqslant 6V\tau + \mathbb{E}\left[\frac{1}{2}\sum_k \sum_{t=1}^{\tau}\left(\check{c}_t^{(k)}(x_t, a_t) + b^2\right)\right] + \frac{4K\tau(1+b)}{VT},$$

$$\leqslant 6V\tau + \frac{K}{2}\left(\frac{1}{T} + B + \tau b^2\right) + \frac{4K\tau(1+b)}{VT}$$

$$\leqslant 6VT + K(1 + B + Tb^2) + \frac{4K(1+b)}{V}$$

$$\leqslant 6VT + 6KB + KTb^2 + 4K\sqrt{T},$$

where the second inequality holds due to the high probability event $\mathcal{E}$, the last inequality holds since $1/V \leqslant \sqrt{T}$ and $b/V \leqslant \sqrt{T}b \leqslant B$, Combine the facts that $Q_1^{(k)} = 0$, $\forall k$ and Cauchy-Schwarz inequality, we have

$$\mathbb{E}\left[\frac{\left(\sum_k Q_{\tau+1}^{(k)}\right)^2}{2K}\right] \leqslant 6VT + 6KB + KTb^2 + 4K\sqrt{T},$$

Rearrange these terms, we obtain that

$$\mathbb{E}\left[\left(\sum_k Q_{\tau+1}^{(k)}\right)^2\right] \leqslant 12KVT + 12K^2B + 2K^2Tb^2 + 8K^2\sqrt{T}.$$

Then we have

$$\mathbb{E}\left[\sum_k Q_{\tau+1}^{(k)}\right] \leqslant 4\sqrt{Kb}T^{\frac{3}{4}} + 4K\sqrt{Tb} + 2Kb\sqrt{T} + 3KT^{\frac{1}{4}}.$$

### B.4.1 CBwK with strict feasibility assumption in Assumption 3

We define the Lyapunov function $\bar{L}_t = \sqrt{\sum_k (Q_t^{(k)})^2} = \|\boldsymbol{Q}_t\|_2$. To establish the expected bound on the virtual queue, we prove conditions (i) and (ii) in Lemma 6 for $\bar{L}_t$. From (17), we have

$$\mathbb{E}[\|\boldsymbol{Q}_{t+1}\|_2^2 - \|\boldsymbol{Q}_t\|_2^2] \leqslant 6V + \mathbb{E}\left[\frac{1}{2}\sum_k \left(\check{c}_t^{(k)}(x_t, a_t) - b\right)^2 \mid \mathcal{H}_t\right] - \sum_k Q^{(k)}\delta b + \frac{4K(1+b)}{VT}$$

$$\leqslant 6V + (1+b)^2 - \sum_k Q^{(k)}\delta b + \frac{4K(1+b)}{VT}$$

$$\leqslant 6V + 6K(1+b)^2 - \sum_k Q^{(k)}\delta b.$$

Given $\mathcal{H}_t = h$ and $\bar{L}_t \geqslant \varphi_t = \frac{4(3V + 3K(1+b)^2)}{\delta b}$, the conditional expected drift of $\bar{L}_t$ is

$$\mathbb{E}[\|\boldsymbol{Q}_{t+1}\|_2 - \|\boldsymbol{Q}_t\|_2 | \mathcal{H}_t = h] \leqslant \frac{1}{2\|\boldsymbol{Q}\|_2}\mathbb{E}[\|\boldsymbol{Q}_{t+1}\|_2^2 - \|\boldsymbol{Q}_t\|_2^2 | \mathcal{H}_t = h]$$

$$\leqslant \frac{6V + 6K(1+b)^2 - \|\boldsymbol{Q}\|_1\delta b}{2\|\boldsymbol{Q}\|_2}$$

$$\leqslant -\frac{\delta b}{2} + \frac{3V + 3K(1+b)^2}{\|\boldsymbol{Q}\|_2}$$

$$\leqslant -\frac{\delta b}{4},$$

where the first inequality holds because $2(\|\boldsymbol{Q}_{t+1}\|_2 - \|\boldsymbol{Q}_t\|_2)\|\boldsymbol{Q}_t\|_2 \leqslant \|\boldsymbol{Q}_{t+1}\|_2^2 - \|\boldsymbol{Q}_t\|_2^2$; the second inequality holds by the "negative drift" in (17) above; the third inequality holds since $\|\boldsymbol{Q}\|_1 \geqslant \|\boldsymbol{Q}\|_2$; and the last inequality holds given the condition $\|\boldsymbol{Q}_t\|_2 \geqslant \varphi_t = \frac{4(3V+3K(1+b)^2)}{\delta b}$. Moreover, for condition (ii) in Lemma 6, we have

$$\|\boldsymbol{Q}_{t+1}\|_2 - \|\boldsymbol{Q}_t\|_2 \leqslant \|\boldsymbol{Q}_{t+1} - \boldsymbol{Q}_t\|_2 \leqslant \|\boldsymbol{Q}_{t+1} - \boldsymbol{Q}_t\|_1 \leqslant K,$$

where the last inequality holds because $|Q_{t+1}^{(k)} - Q_t^{(k)}| \leqslant 1 - b, \ \forall k \in [K]$.

Let $\rho = \frac{\delta b}{4}$, $\nu_{\max} = K$ and recall $Q_1^{(k)} = 0, \ \forall k \in [K]$. We are ready to apply Lemma 6 for $\bar{L}(t)$ and obtain

$$\mathbb{E}\left[e^{\zeta\|\boldsymbol{Q}_t\|_2}\right] \leqslant 1 + \frac{2e^{\zeta(\nu_{\max}+\varphi_t)}}{\zeta\rho} \text{ with } \zeta = \frac{\rho}{\nu_{\max}^2 + \nu_{\max}\rho/3}. \tag{18}$$

Then according to Cauchy-Schwarz inequality, we have

$$\mathbb{E}\left[e^{\frac{\zeta}{\sqrt{K}}\|\boldsymbol{Q}_t\|_1}\right] \leqslant 1 + \frac{2e^{\zeta(\nu_{\max}+\varphi_t)}}{\zeta\rho},$$

Applying Jensen's inequality, we obtain that

$$e^{\frac{\zeta}{\sqrt{K}}\mathbb{E}[\|\boldsymbol{Q}_t\|_1]} \leqslant \mathbb{E}\left[e^{\frac{\zeta}{\sqrt{K}}\|\boldsymbol{Q}_t\|_1}\right] \leqslant 1 + \frac{2e^{\zeta(\nu_{\max}+\varphi_t)}}{\zeta\rho},$$

Finally, for any $t \in [T]$

$$\begin{aligned}
\mathbb{E}\left[\sum_k Q_t^{(k)}\right] &\leqslant \frac{\sqrt{K}}{\zeta}\log\left(1 + \frac{2e^{\zeta(\nu_{\max}+\varphi_t)}}{\zeta\rho}\right)\\
&\leqslant \frac{\sqrt{K}}{\zeta}\log\left(1 + \frac{8\nu_{\max}^2 e^{\zeta(\nu_{\max}+\varphi_t)}}{3\rho^2}\right)\\
&\leqslant \frac{\sqrt{K}}{\zeta}\log\left(\frac{11\nu_{\max}^2 e^{\zeta(\nu_{\max}+\varphi_t)}}{3\rho^2}\right)\\
&\leqslant \frac{2\sqrt{K}}{\zeta}\log\left(2\nu_{\max}/\rho\right) + K^{\frac{3}{2}} + \sqrt{K}\varphi_t\\
&= \frac{2\sqrt{K}}{\zeta}\log\left(8K/\delta b\right) + K^{\frac{3}{2}} + \frac{4\sqrt{K}(3V + 3K(1+b)^2)}{\delta b}\\
&= \frac{2\sqrt{K}}{\zeta}\log\left(8K/\delta b\right) + K^{\frac{3}{2}} + \frac{12K^2(1+b)^2}{\delta b} + \frac{12\sqrt{KT}}{\delta},
\end{aligned}$$

where the second, the third and the fourth inequality comes from the definition of $\zeta$ and the fact that $0 < \rho \leqslant \nu_{\max}$ and $\nu_{max} = K$, the last two equalities comes from the definition of $\varphi_t$ and $V$.

## B.5 Proof of Lemma 5 under Algorithm 1

Recall the definition of virtual stopping time

$$\tau_0 = \operatorname{argmin}_{\tau'\in[T]}\left\{\tau' \mid Q_{\tau'+1}^{(k)} + b\tau' + M_{\tau'}^{(k)} \geqslant B, \exists k\right\},$$

where $M_\tau^{(k)} = \sum_{t=1}^{\tau}(c^{(k)}(x_t, a_t) - \check{c}_t^{(k)}(x_t, a_t))$. Divide both sides of the stopping time definition inequality by $b$ and take the expectation, we have

$$\mathbb{E}[T - \tau] \leqslant \mathbb{E}[T - \tau_0]$$

$$\leqslant \mathbb{E}\left[Q_{\tau+1}^{(k')}/b\right] + \mathbb{E}[M_\tau^{(k')}/b]$$

$$\leqslant \mathbb{E}\left[\|\boldsymbol{Q}_{\tau+1}\|_1/b\right] + \mathbb{E}[M_\tau^{(k')}/b],$$

where the first inequality holds since $\tau \leqslant \tau_0$ according to the definition (8). From Assumption 2, for any $k$

$$M_\tau^{(k)} \leqslant 2\sum_{t=1}^{\tau} \varepsilon_t(x_t, a_t, p) \leqslant 2\sum_{t=1}^{T} \varepsilon_t(x_t, a_t, p) \leqslant 2C_0\sqrt{T}\log(T/p),$$

then the stopping time bound can be established by

$$\mathbb{E}\left[T - \tau\right] \leqslant \mathbb{E}[\|\boldsymbol{Q}_{\tau+1}\|_1/b] + \frac{2C_0\sqrt{T}\log(T/p)}{b}$$

We first consider the case without strict feasibility assumption, then we can establish the bound through Lemma 4 such that:

$$\mathbb{E}\left[T - \tau\right] \leqslant 4\sqrt{\frac{K}{b}}T^{\frac{3}{4}} + 4K\sqrt{\frac{T}{b}} + 2K\sqrt{T} + \frac{3KT^{\frac{1}{4}}}{b} + \frac{2C_0\sqrt{T}\log(T/p)}{b}.$$

We can then get a refined result with the Slater condition in Assumption 3. Since we have established an upper bound in Lemma 4, the stopping time can be bounded by

$$\mathbb{E}\left[T - \tau\right] \leqslant \frac{2\sqrt{K}}{\zeta b}\log\left(8K/\delta b\right) + K^{\frac{3}{2}}b^{-1} + \frac{12K^2(1+b)^2}{\delta b^2} + \frac{12\sqrt{KT}}{\delta b} + \frac{2C_0\sqrt{T}\log(T/p)}{b}.$$

### B.6 PROOF OF THEOREM 1 UNDER ALGORITHM 1

Now we aggregate the regret after stopping and before stopping as follows. For the general case, we have

$$\text{Regret}(T) \leqslant \underbrace{\nu^*\mathbb{E}[T-\tau]}_{\text{regret after stopping}} + \underbrace{\mathbb{E}\left[\sum_{t=1}^{\tau} r(x_t, a_t^*)\right] - \mathbb{E}\left[\sum_{t=1}^{\tau} r(c_t, a_t)\right]}_{\text{regret before stopping}}$$

$$\leqslant C_1 + 2C_0\sqrt{T}\log(T/p) + K(2+b)\sqrt{T}$$

$$+ \left(4\sqrt{\frac{K}{b}}T^{\frac{3}{4}} + 4K\sqrt{\frac{T}{b}} + 2K\sqrt{T} + \frac{3KT^{\frac{1}{4}}}{b} + \frac{2C_0\sqrt{T}\log(T/p)}{b}\right)\nu^*. \quad (19)$$

With the Slater condition in Assumption 3, we have

$$\text{Regret}(T) \leqslant \underbrace{\nu^*\mathbb{E}[T-\tau]}_{\text{regret after stopping}} + \underbrace{\mathbb{E}\left[\sum_{t=1}^{\tau} r(x_t, a_t^*)\right] - \mathbb{E}\left[\sum_{t=1}^{\tau} r(c_t, a_t)\right]}_{\text{regret before stopping}}$$

$$\leqslant C_1 + 2C_0\sqrt{T}\log(T/p) + K(2+b)\sqrt{T}$$

$$+ \left(\frac{2\sqrt{K}}{\zeta b}\log\left(8K/\delta b\right) + K^{\frac{3}{2}}b^{-1} + \frac{12K^2(1+b)^2}{\delta b^2} + \frac{12\sqrt{KT}}{\delta b} + \frac{2C_0\sqrt{T}\log(T/p)}{b}\right)\nu^*.$$

$$(20)$$

which proves the regret bounds in Theorem 1.

## C SUPPORTING LEMMAS

### C.1 LYAPUNOV DRIFT LEMMA

We present a lemma that will be used to derive the high probability of $\{Q_t\}$. The lemma is from Liu et al. (2021), which is a minor variation of Lemma 4.1 Neely (2016; 2022) and the results in Hajek (1982), where the radius of $\varphi_t$ could be time dependent.

**Lemma 6** *Let $S(t)$ be a random process, $\Phi(t)$ be its Lyapunov function with $\Phi(0) = \Phi_0$ and $\Delta(t) = \Phi(t+1) - \Phi(t)$ be the Lyapunov drift. Given an increasing sequence $\{\varphi_t\}$, $\rho$ and $\nu_{\max}$ with $0 < \rho \leqslant \nu_{\max}$, if the expected drift $\mathbb{E}[\Delta(t)|S(t) = s]$ satisfies the following conditions:*

*(i) There exists constants $\rho > 0$ and $\varphi_t > 0$ such that $\mathbb{E}[\Delta(t)|S(t) = s] \leqslant -\rho$ when $\Phi(t) \geqslant \varphi_t$, and*

*(ii) $|\Phi(t+1) - \Phi(t)| \leqslant \nu_{\max}$ holds with probability one;*

*then we have*

$$\mathbb{E}[e^{\zeta\Phi(t)}] \leqslant e^{\zeta\Phi_0} + \frac{2e^{\zeta(\nu_{\max}+\varphi_t)}}{\zeta\rho}, \tag{21}$$

*where $\zeta = \frac{\rho}{\nu_{\max}^2 + \nu_{\max}\rho/3}$.*

## D    EXPERIMENTAL DETAILS AND ADDITIONAL EXPERIMENTS

### D.1    EXPERIMENTAL DETAILS

In this section, we provide the experimental details of our evaluations. We first discuss the learning-to-rank dataset we use as the reward functions: **Microsoft Learning to Rank.** We use the MSLR-WEB30K dataset Qin & Liu (2013) which is available at `https://www.microsoft.com/en-us/research/project/mslr/`. This dataset has 31,278 arrivals, and the contextual dimension is 136. We extract $|\mathcal{A}| = 20$ documents (arms) per query and the reward is defined as the relevance judgments in the dataset which take 5 values from 0 (irrelevant) to 4 (perfectly relevant). We set the time horizon $T = 5000$ and randomly draw an arrival from the 31,278 data points.

We then provide the hyperparameters of tested algorithms in our experiments.

| Algorithm | Parameters |
|---|---|
| AUPD Algorithm | $V = 0.1b\sqrt{T}$ |
| PGD Adaptive Chzhen et al. (2024) | $M_T = 4\sqrt{T} + 2\sqrt{T\log T},\ \delta b = 1/\sqrt{T}$ |
| SquareCBwK Han et al. (2023) | $U = \sqrt{T\log T},\ \gamma = 0.1\sqrt{T/U}$ |

Table 2: Parameters in the experiment.

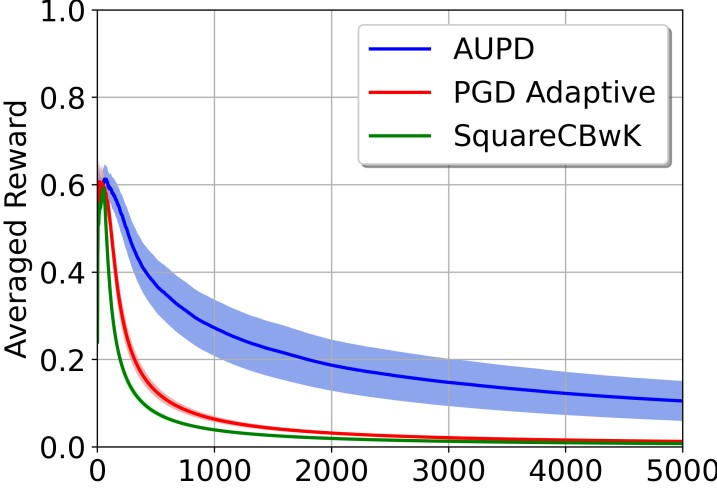

Figure 2: Budget $B = 30$.

Table 3: Average cumulative rewards under various budgets.

|  | B = 30 | B = 100 | B = 600 | B = 1000 |
|---|---|---|---|---|
| Our Algorithm (AUPD) | 0.105 | 0.233 | 0.238 | 0.344 |
| PGD Adaptive | 0.012 | 0.051 | 0.230 | 0.336 |
| SquareCBwK | 0.007 | 0.028 | 0.174 | 0.298 |

## D.2 EXPERIMENTS UNDER OTHER BUDGET REGIMES

Recall the previous experiments with budgets $B = \{100, 600, 1000\}$ to represent different budget regimes $\{\Theta(\sqrt{T}), \Theta(T^{3/4}), \Theta(T)\}$. We consider the same setting and vary the budget with smaller budgets, where $B = \{30\}$. The experimental results are shown in Figure 2, which suggests that our algorithm adapts effectively to varying budget regimes and achieves much better performance as the budget decreases or the constraints become tight. A summary of average cumulative rewards can be found in Table 3, where the values for $B = \{100, 600, 1000\}$ are from Figure 1.

