# OpenReview forum: "On Stochastic Contextual Bandits with Knapsacks in Small Budget Regime"
_ICLR.cc/2025/Conference — ICLR 2025 Poster_

### Official Review · Reviewer_E8sG · 2024-11-04

**Soundness:** 3
**Presentation:** 2
**Contribution:** 2
**Rating:** 5
**Confidence:** 3

**Summary:**

The paper studies stochastic contextual bandits with knapsack. In this model, at each round the learner observes a context before taking an action. The goal of the learner is to maximize its utility subject to a knapsack constraint. The paper presents an algorithm that guarantees an instance-dependent regret bound both with and without the strictly feasibility assumption. The paper provides meaningful guarantees even with budget $B=\Omega(\sqrt{T})$.

**Strengths:**

It is the first paper to work without knowing the safe margin or without a safe margin.

**Weaknesses:**

The improvements with respect to previous works are minimal, and it is not clear the importance and technical challenges in removing the assumptions in previous works. Indeed, the assumption of a cost 0 "do nothing" action is fairly natural.

**Questions:**

The algorithm and the analysis look quite standard. Which are the technical difference and challenges with respect to previous works? Why is Lyapunov Drift more effective than previous approaches?

---

> ### Author Response · Authors · 2024-11-19
> **Response to Reviewer E8sG**
>
> We appreciate the reviewer’s comments and want to address your major concerns below.
>
> - **The CBwK setting without a "null action" is relevant and common in most practical settings.**
>   The "null action" is not always feasible in most practical applications. We list a few representative examples:
>   - In a **patient boarding system** [R1, R2], patients arrive sequentially, and hospitals must allocate suitable medical resources (e.g., physicians or treatment facilities) to each patient. Here, a "null action" or "do nothing action" is neither suitable nor practical.
>   - In a **load-balancing system** for a cloud platform [R3], user-submitted jobs (e.g., machine learning workloads) are distributed to servers for processing. Assigning a "null server" with zero rewards and costs is not a good choice for either users or the platform, as jobs that fail to find an available server upon arrival are blocked and ultimately lost.
>   - In a **recommendation platform** [R4], the platform must display appropriate items to each incoming user to maximize click-through rates, where item display typically incurs a cost. A "null item" is not a good option as it would degrade both the user experience and platform profits.
>
>   We will incorporate these examples in our revision if the reviewer considers them appropriate.
>
> - **Technical challenges and differences with respect to existing studies.**
>   CBwK is a challenging problem as it requires balancing reward maximization and resource consumption without prior knowledge of the context distribution (the "spend-or-save dilemma"). This challenge is particularly pronounced in settings with small budgets and without strict feasibility.
>
>   Previous studies, such as Han et al. (2023), Slivkins et al. (2023), and Chzhen et al. (2024), rely on knowledge of the strict feasibility margin and require either an extra learning process to estimate the optimal value $\nu^*$ or a doubling trick to learn the optimal step-size for dual updates (a proxy for managing budget usage).
>
>   Without the strict feasibility assumption or feasibility margin knowledge, it is unclear how these approaches address the "spend-or-save dilemma."
>
>   In contrast, our algorithm is **direct, single-stage, and adaptive**, leveraging the initial budget information in both the primal decision domain and the dual domain through the virtual queue design. This eliminates the need for extra estimation or tuning processes, enabling effective operation even without relying on the strict feasibility assumption.
>
> - **The advantage of Lyapunov drift analysis.**
>   The budget consumption process is key to understanding the regret performance of CBwK. Lyapunov drift analysis is effective in analyzing this process from two perspectives:
>   1. With the **strict feasibility assumption**, Lyapunov drift analysis establishes an upper bound on the virtual queue (a proxy for the budget consumption process) without requiring explicit knowledge of the strict feasibility margin. This sets our approach apart from existing methods, where no-regret learning techniques (e.g., Slivkins et al., 2023) and optimization-based techniques (e.g., Han et al., 2023; Chzhen et al., 2024) rely on explicit feasibility margin knowledge to bound dual updates (also proxies for the budget consumption process).
>   2. Without the **strict feasibility assumption**, Lyapunov drift analysis, particularly using quadratic Lyapunov functions, can still provide an upper bound on the virtual queue. This might not be feasible with existing techniques, such as those by Slivkins et al. (2023), Han et al. (2023), and Chzhen et al. (2024), where the strict feasibility assumption is a requirement.
>
> We hope that our response addresses the reviewer’s concerns and that the reviewer can re-evaluate our work. Please let us know if you have any further comments, and we will try our best to address them.
>
> ---
>
> ### References
>
> - [R1] Zhalechian, M., Keyvanshokooh, E., Shi, C., et al. *Personalized hospital admission control: A contextual learning approach*. Available at SSRN, 2020.
> - [R2] Tewari, A., Murphy, S. A. *From ads to interventions: Contextual bandits in mobile health*. *Mobile Health: Sensors, Analytic Methods, and Applications*, 2017.
> - [R3] Verma, A., Pedrosa, L., Korupolu, M., et al. *Large-scale cluster management at Google with Borg*. In *Proceedings of European Conference on Computer Systems*, 2015.
> - [R4] Smith, B., Linden, G. *Two decades of recommender systems at Amazon.com*. *IEEE Internet Computing*, 2017.

---

> > ### Author Response · Authors · 2024-11-26
> > **Response to Reviewer E8sG**
> >
> > We sincerely appreciate your insightful comments and suggestions, which have greatly contributed to improving the quality of our work. We hope our response has addressed your concerns. If you have any further questions, please let us know so we can address them before the rebuttal phase ends. Thank you very much for your time!

---

### Official Review · Reviewer_2HF3 · 2024-11-04

**Soundness:** 4
**Presentation:** 3
**Contribution:** 3
**Rating:** 8
**Confidence:** 5

**Summary:**

This paper studies the contextual bandits with knapsacks problem with a focus on when the budget is "small" (i.e., $B = \Omega(\sqrt{T})$). The authors present an algorithm with $\sqrt{T}$ regret under the strict feasibility assumption and $T^{\frac{3}{4}}$ regret without it. Notably, the algorithm does not need to know which of these two regimes it is in. The algorithm improves upon prior work by being a single-stage algorithm and by using the cumulative over-consumption of resources as a Lagrange multiplier.

**Strengths:**

* Contextual bandits with knapsacks (CBwK) is an important model of online decision-making with many applications. This paper advances our knowledge in this area by improving upon an important limitation in existing work, namely, considering the regime of $B = \Omega(\sqrt{T})$ with and without strict feasibility. Their regret bound for the setting without strict feasibility is also the first such result.
* The algorithm has some good features: it is simple and it does not need to know whether strict feasibility holds or not.
* The choice to use the cumulative over-consumption as a Lagrange multiplier is a natural and effective idea.

**Weaknesses:**

* Overall, it's a well-written paper. But I would have loved to see some intuition behind the proofs of the lemmas in the main text. When I read through the proofs in the appendix, I followed the steps. But I still would have liked to see an explanation in words about the main ideas in the proof.
* In the three settings considered in the experiments, the proposed algorithm is strictly better than the alternatives in only 1 setting; it is roughly the same as PGD Adaptive in the other two settings. It would have been nice to see other settings in which the proposed algorithm is strictly better than the alternatives.

**Questions:**

Questions:
1. You claim that your algorithm is "budget-aware" because of the parameter $V = b \sqrt{T}$, as opposed to $V = \sqrt{T}$. I am wondering how crucial this choice is. Quantitatively, what would have been the regret bound had you chosen $V = \sqrt{T}$? Qualitatively, where does including $b$ in the parameter help you in the proof?
2. Why are the variables $Q_t^k$ called "virtual queues"? I'm trying to understand the motivation for this terminology and if there is a connection to literature on "virtual queues" that is helpful here.
3. I understand the technical details proof of Lemma 5, but can you provide some intuition for how Lemma 4 is helpful for proving Lemma 5?
4. Do you have thoughts on whether $T^{\frac{3}{4}}$ can be improved when strict feasibility does not hold?

Minor comments and typos:
1. Line 314: "is not necessary to hold" -> "does not necessarily hold".
2. Line 337: "provide detailed proof" -> "provide a detailed proof".
3. Equation 8: This is a very minor nitpick, but I personally find it easier to parse the statement when it's written in the format $\exists k$ s.t. {condition}.
4. Line 346: Should this be denoted $a^*_t$ instead of $a^*$ since a different action will be sampled in different rounds depending on the context? Also, "be the optimal action sampling from it" -> "optimal action sampled from it".
5. Line 363-364: What is $f$? Did you mean $r$?
6. Line 368: Regret($x_t, a$) has not been defined before. But I guess it means $r(x_t, a^*_t) - r(x_t, a)$?
7. Line 400-401: "against the average usage $t \times b$ for the round t" - Isn't the average usage $b$?
8. Line 418-419: This is a very minor nit, but I suggest rewording "we have established" to "we establish". When I first read this, I was wondering where this established in the paper so far. Then I read the full sentence and realized it is proved in the next lemma.
9. Line 483-484: Did you mean 1a instead of 1b?
10. Line 708-709: Should $c$ be $\check{c}$?
11. Line 981-982: It took me some time to understand what you meant by "divide both sides". It might be clearer to explicitly say that you divide both sides of the inequality inside the argmin.
12. In Section B.3.1, you use Assumption 3. Then you use the resulting inequality (16) in Section B.4 where you don't assume Assumption 3. My guess is that this is ok since Eq 16 has a $-Q \delta b$ term and you upper bound this by 0 in Section B.4? It might be good to be clear about this in Section B.3.1.

---

> ### Author Response · Authors · 2024-11-19
> **Response to Reviewer 2HF3**
>
> We sincerely appreciate the reviewer’s great comments and positive evaluation of our paper. We will provide more intuition on the theoretical analysis and fix the typos in our revision. We focus on addressing your major comments as follows.
>
> - **Experiments:**
>   The experiments in Figure 1 of our submission suggest that our algorithm performs particularly well as the budget becomes small. We consider the same setup as in the paper but with an even smaller budget, $B = 30$, to simulate the regime $B = \Theta(T^{1/4})$, even though our theoretical results are guaranteed only for $B = \Theta(\sqrt{T})$. Recall the previous experiments with budgets $B = \{100, 600, 1000\}$ represent the budget regimes $\{\Theta(\sqrt{T}), \Theta(T^{3/4}), \Theta(T)\}$. The average cumulative rewards are summarized in the table below, where the values for $B = \{100, 600, 1000\}$ are from Figure 1. These results suggest that our algorithm adapts effectively to varying budget regimes and achieves much better performance as the budget decreases or the constraints become tight. We will include these additional results in our revision per the reviewer’s suggestion.
>
> |                 | B = 30 | B = 100 | B = 600 | B = 1000 |
> |-----------------|--------|---------|---------|----------|
> | Our Algorithm (AUPD) | 0.105  | 0.233   | 0.238   | 0.344    |
> | PGD Adaptive    | 0.012  | 0.051   | 0.230   | 0.336    |
> | SquareCBwK      | 0.007  | 0.028   | 0.174   | 0.298    |
>
> - **The budget-aware design $V$:**
> The trade-off parameter $V = b\sqrt{T}$ is budget-aware and adaptive to different budget regimes. Qualitatively, the budget-aware design of $V = b\sqrt{T}$ prompts more conservative actions, ensuring the resource is spent more carefully in the small budget regime. Removing $b$ in $V$ (i.e., $V = \sqrt{T}$) might cause the algorithm to overuse the resource, especially when the budget is small. This overuse could result in a large virtual queue, causing our algorithm to stop early and incur a large regret. Quantitatively, this is reflected in the proof of Lemma 4, where we establish the upper bound of the virtual queue. When $V = \sqrt{T}$, we would have a large virtual queue $O(\sqrt{T}/b)$ (see lines 953–969 in Section B.4.1) instead of $O(\sqrt{T})$ obtained with the budget-aware design. This eventually leads to a worse regret bound of $O((1 + \frac{\nu^*}{b^2})\sqrt{T})$ compared to the $O((1 + \frac{\nu^*}{b})\sqrt{T})$ bound with the budget-aware design.
>
> - **The terminology of "virtual queue":**
>   The concept of a virtual queue originates from queueing theory and is widely used in networking and operations research [R1, R2, R3]. In a real queueing system, customers arrive, receive service, and leave, with the queue capturing the carryover effect and representing the number of waiting customers. In CBwK, the term $Q_t$ represents the cumulative overuse of a resource, where the “arrival” corresponds to the current resource consumption and the “service” corresponds to the average budget. This analogy to real queue dynamics motivates the term “virtual queue.”
>
> - **Intuition behind Lemma 4 and Lemma 5:**
>   Lemma 4 establishes the upper bound of the virtual queue $Q_t$, which quantifies how much of the budget is overused until round $t$. Intuitively, the algorithm would stop when the resource is used up, i.e., the actual resource usage $\sum_{t=1}^{\tau} c_t \geq Q_{\tau} + \tau b \geq B$. We hope $Q_t$ is small so that our algorithm does not stop early. Lemma 4 suggests this is true, and $Q_t = O(\sqrt{T})$ holds. This translates into a lower bound on stopping time in Lemma 5.
>
> - **Potential improvements on $T^{3/4}$:**
> The main challenge in improving the $T^{3/4}$ regret lies in obtaining a refined bound on the virtual queue when the strict feasibility does not hold, as this directly impacts the stopping time and the regret. To address this, we may need to redesign the budget-aware action and the virtual queue update to make the algorithm schedule the resource more effectively. From an analytical perspective, we may need to develop new Lyapunov functions that better capture the overused resource so that we can achieve a refined upper bound on it.
>
> - **Minor comments and typos:**
>   1, 2, 3, 4, 5, 7, 8, 9, 10, 11: We greatly appreciate your comments. We will carefully modify them in our revision.
>   6: Apologies for the confusion. The definition of $\text{Regret}(x_t, a)$ first appears in Appendix, line 697. We will move it up to an earlier section where it is first referenced.
>   12: Thank you for pointing out this mismatch. We cannot directly use (16). The correct way is to analyze the drift function based on the equation on line 848 by taking $a \sim \pi^*$. We will revise it.

---

> > ### Author Response · Authors · 2024-11-19
> > **Reference**
> >
> > - [R1] Neely, M. *Stochastic network optimization with application to communication and queueing systems*. Springer Nature, 2022.
> > - [R2] Hajek, B. *Hitting-time and occupation-time bounds implied by drift analysis with applications*. Advances in Applied Probability, 1982.
> > - [R3] Eryilmaz, A., Srikant, R. *Asymptotically tight steady-state queue length bounds implied by drift conditions*. Queueing Systems, 2012.

---

> > > ### Author Response · Authors · 2024-11-26
> > > **Response to Reviewer 2HF3**
> > >
> > > We sincerely appreciate your acknowledgment, positive feedback, and insightful suggestions, which have greatly helped us improve our work! If you have any further concerns or questions, please don't hesitate to let us know.

---

> > > > ### Comment · Reviewer_2HF3 · 2024-11-26
> > > >
> > > > Thank you for the response and the clarifications!

---

### Official Review · Reviewer_UqkS · 2024-11-05

**Soundness:** 2
**Presentation:** 2
**Contribution:** 2
**Rating:** 6
**Confidence:** 2

**Summary:**

The paper studies contextual bandit with knapsack in the small budget regime (B=o(T), B=\Omega(T)).
The paper provides an algorithm which achieves O(sqrt(T)/(d*B/T)) without knowing "slater" action d and O(T^{3/4}/sqrt(B/T)) without strictly feasibility (d=0).

**Strengths:**

The setting studied is interesting and not trivial and leads to intriguing scientific questions.

**Weaknesses:**

The paper is a bit difficult to read, and I think I could benefit from being inserted in a larger context with respect to prior/concurrent literature.
For example, the recent literature that considers bandits with constraints (of which bwk is a special case) has much more relevancy than what the authors seem to realize. In particular, I'm referring to

[1] Raunak Kumar and Robert Kleinberg. Non-monotonic resource utilization in the bandits with
knapsacks problem. In Advances in Neural Information Processing Systems (NeurIPS), 2022.
[2] Bernasconi, Martino, Matteo Castiglioni, and Andrea Celli. "No-Regret is not enough! Bandits with General Constraints through Adaptive Regret Minimization."
[3] Slivkins, Aleksandrs, Karthik Abinav Sankararaman, and Dylan J. Foster. "Contextual bandits with packing and covering constraints: A modular lagrangian approach via regression." The Thirty Sixth Annual Conference on Learning Theory. PMLR, 2023.
[4] Bernasconi, Martino, et al. "Bandits with Replenishable Knapsacks: the Best of both Worlds." The Twelfth International Conference on Learning Representations.

**Questions:**

What are the technical challenges that prevent adapting existing techniques such as "Contextual bandits with packing and covering constraints: A modular lagrangian approach via regression" to this context? Is it true that the authors assume large budget, but it is not obvious that their framework cannot be used to solve the small budget case. At least this merits an explanation, which might also add relevancy to your technical contributions, which at the moment are not really highlighted.
When are your results meaningful? For example, if B=sqrt(T), then the results are linear and meaningless in the case of no strictly feasible. It would be helpful to plot/discuss a B=T^alpha vs R_T tradeoff as a function of alpha.
How do your algorithms really solves the small budget? I fail to understand how your algorithm behaves differently when B=sqrt(T) or B=\Theta(T).
What would happen if you used one of the existing algorithms that do not use knowledge of the later parameter and applied it to the small-budget case?

---

> ### Author Response · Authors · 2024-11-19
> **Response to Reviewer UqkS**
>
> We very much appreciate the reviewer for the constructive comments and want to address your major concerns below.
>
> - **Related works (citation numbers are from reviewer comments):**
>   Thank you for pointing out these related papers. We will definitely incorporate them in our revision and provide a more comprehensive overview. Specifically, we will discuss them from the perspectives of model, algorithm design, and theoretical results. For example:
>   - [1] and [4] study non-monotonic/replenishable resource utilization in non-contextual bandits with knapsacks.
>   - [2] studied both stochastic and adversarial bandits with constraints, where constraint violations are allowed. Adapting their algorithm to the hard-stopping setting might require additional procedures.
>   - [3] was included in our initial submission, but we will provide a more detailed discussion and comparison (e.g., the strict feasibility assumption) to highlight our contribution, as you suggested.
>
> - **Technical challenges of the existing approaches:**
>   For CBwK (i.e., the hard-stopping setting in Corollary 5.3(c)), Slivkins et al. (2023) assume the large budget regime, which is more explicitly presented in their standalone technical report focusing on CBwK ([arXiv:2211.07484v1](https://arxiv.org/abs/2211.07484v1)). Slivkins et al. (2023) inherited the technique from Immorlica et al. (2022), where a large budget and strict feasibility margin are assumed.
>   It remains unclear whether their technique can be extended to small-budget settings without knowledge of a strict feasibility margin or even without assuming strict feasibility.
>
>   Other existing techniques in Han et al. (2023) and Chzhen et al. (2024) have similar issues, relying on the knowledge of strict feasibility margins to determine the key trade-off parameter (e.g., the dual variables). Without this knowledge, applying these algorithms may fail to achieve a good or sublinear regret, as it is challenging to schedule resources effectively without such key information.
>
>   However, our algorithm only uses the initial budget information to adapt to the (small) budget regime and does not require knowledge of the strict feasibility margin or even the assumption of strict feasibility.
>
> - **Our results are meaningful in most typical and practical settings:**
>   Recall the regret is $O(\sqrt{T} + \frac{\nu^*}{\sqrt{b}} T^{3/4})$ without strict feasibility assumption, where the parameter $\frac{\nu^*}{\sqrt{b}}$ can be understood as a problem-dependent parameter. The typical and practical setting would have $\nu^* = \Theta(b)$ (i.e., $T\nu^* = \Theta(B)$), representing "one unit of reward earned by consuming one unit of cost." In this setting, the dominant term becomes $\frac{\nu^*}{\sqrt{b}} T^{3/4} = \sqrt{b}T^{3/4},$ which is meaningful when $B = \Omega(\sqrt{T})$.
>
>   In an "extreme" setting where $T\nu^* = \Theta(T)$ is earned with little cost consumption ($B = \Omega(\sqrt{T})$), i.e., "one unit of reward is earned with very little ($1/\sqrt{T}$) effort," a linear regret may be unavoidable. The lower bound $\Omega((1 + \nu^*/b) \sqrt{T})$ in Chzhen et al. (2024) under the strict feasibility assumption suggests this lower bound is $\Omega(T)$ in the "extreme" setting.
>
>   We will incorporate these discussions in our revision by explicitly representing the regret with respect to $\alpha$.
>
> - **Our adaptive design "solves" the small budget:**
>   When the budget varies from $\Theta(\sqrt{T})$ to $\Theta(T)$, our algorithm adapts to different budget regimes through the carefully designed trade-off parameter $V = b\sqrt{T}$ and the virtual queue updates. Intuitively, when the budget becomes smaller, $V$ prompts more conservative actions, and the budget is spent more carefully.
>
>   With this adaptive design and refined theoretical analysis, we achieve a regret of $O(\sqrt{T} + \frac{\nu^*}{\delta b} \sqrt{T})$ with $B = \Omega(\sqrt{T})$, which is consistent with the result in the large budget regime ($\Theta(T)$) and matches the lower bound suggested in Chzhen et al. (2024). Without the strict feasibility assumption, our algorithm establishes a regret bound of $O(\sqrt{T} + \frac{\nu^*}{\sqrt{b}} T^{3/4})$.
>
>   These bounds are meaningful in typical settings, as discussed above. We believe these results provide a relatively complete picture of BwK in the small budget regime.

---

> > ### Author Response · Authors · 2024-11-26
> > **Response to Reviewer UqkS**
> >
> > We sincerely appreciate your insightful comments and valuable suggestions, which have greatly improved our work quality. We hope our responses adequately address your concerns. If you have any further questions, please feel free to let us know, and we will be happy to address them before the rebuttal phase concludes.

---

### Official Review · Reviewer_Nbbq · 2024-11-09

**Soundness:** 3
**Presentation:** 3
**Contribution:** 3
**Rating:** 8
**Confidence:** 2

**Summary:**

This paper studies the contextual bandits with knapsack problem under the small budget scenario. Previous research needs to know the safety margin of the budget constraint for their algorithms. This work gives an algorithm achieving the best known regret without knowledge of the safety margin. Furthermore, the algorithm can achieve sub-linear regret with not-strictly-feasible constraints in some cases.

**Strengths:**

This paper gives a good contribution to a natural problem. The contextual bandit with small budget knapsacks is a natural problem. Existing research requires knowledge about the safety margin, which is hard to know. This paper gets rid of this demanding requirement. Furthermore, this paper gives the first result in the not-strictly-feasible case.

This paper is very well written. It is really easy to follow the paper. Everything is clearly defined and stated.

Overall, I think this is a well written paper, makes a good contribution to a natural problem. I am happy to see it published.

**Weaknesses:**

This paper has few weaknesses; I would only suggest that the authors further discuss the relationship between contextual bandits with knapsack and standard bandits with knapsack, and provide the existing lower bound. Please see the specific questions below.

**Questions:**

Question 1: What is the relationship between contextual bandits with knapsack and standard bandits with knapsack? Although we are working with contextual bandits, the final result does not rely on the context numbers. Is this due to the learning oracle? Given a learning oracle, are there any technical differences between this approach and the algorithms for standard bandits with knapsack based on UCB and primal-dual methods?

Question 2: What is the existing lower bound for this problem? How large is the gap between the lower bound and the existing upper bound? Please discuss this point.

---

> ### Author Response · Authors · 2024-11-19
> **Response to Reviewer Nbbq**
>
> We sincerely thank the reviewer for the encouraging comments. We would like to address your questions as follows (citations are consistent with our submission).
>
> - **BwK and Contextual BwK**:
>   The standard bandits with knapsacks (BwK) is a special case of Contextual BwK (CBwK), where no context exists, or a single/fixed context exists. CBwK is usually much more challenging because it needs to balance reward maximization and resource consumption without prior knowledge of context distribution.
>
>   Our final results depend not on the number of contexts but on the context dimension (we apologize for not explicitly clarifying this relationship).
>   This is due to two main reasons: 1) The use of learning oracles, where the learning error depends on the context dimension, and more importantly. 2) The budget-aware/adaptive design, which implicitly learns the context distribution and takes "greedy" decisions to guarantee "context-number-free" dependence.
>
> - **Technical difference with existing primal-dual approaches**:
>   Existing primal-dual approaches with learning oracles, such as those in Agrawal & Devanur (2016), Han et al. (2023), and Chzhen et al. (2024), rely on prior knowledge of the strict feasibility margin and often require additional steps, such as estimating the optimal value $\nu^*$ through an extra learning process or employing a doubling-trick to learn the optimal step size for dual gradient descent. These steps and information are essential for effectively utilizing resources to maximize rewards.
>
>   However, our algorithm is direct, single-stage, and adaptive. It leverages the initial budget information only in the primal decision domain and in the dual domain through the virtual queue design. This eliminates the need for extra estimation or tuning processes, allowing our algorithm to operate effectively even without the strict feasibility assumption.
>
> - **Lower Bound in CBwK**:
>   The lower bound in CBwK is relatively rare in the literature. The classical lower bound for CBwK is $\Omega(\sqrt{T})$ in Agrawal & Devanur (2016), derived by reducing the problem into unconstrained contextual bandits. However, this lower bound did not capture the effect of knapsack constraints. To our knowledge, the most relevant lower bound for CBwK is from Chzhen et al. (2024). With the assumption of strict feasibility, Section 4 (or Section E) in Chzhen et al. (2024) provides a problem-dependent lower bound of $\Omega((1+ \|\boldsymbol{\lambda}_b^*\|) \sqrt{T}),$ where $\boldsymbol{\lambda}_b^*$ is the optimal dual variable with the average budget $b.$   This result implies the lower bound is $\Omega((1 + \nu^*/b) \sqrt{T})$ for CBwK according to the duality of linear programming (LP) formulation.
>   Therefore, our regret bound is tight when the assumption of strict feasibility holds. However, no existing lower bounds are reported without the assumption of strict feasibility, which is a very interesting future work.

---

> > ### Author Response · Authors · 2024-11-26
> > **Response to Reviewer Nbbq**
> >
> > Thank you very much for your acknowledgment and your positive feedback on our work!  If you have any further questions, please don't hesitate to let us know.

---

> ### Comment · Reviewer_Nbbq · 2024-11-27
> **Clarification on My Questions**
>
> Dear Authors,
>
> It seems that you have misunderstood my question:
> "Given a learning oracle, are there any technical differences between this approach and the algorithms for standard bandits with knapsack based on UCB and primal-dual methods?"
>
> Your response addressed the relationship between your algorithm and other primal-dual approaches that rely on a learning oracle. However, that’s not what I meant. My question refers to the many algorithms designed for BwK (the non-contextual version), many of which are based on UCB and primal-dual methods.
>
> What I want to understand is: apart from relying on a learning oracle, what are the differences between algorithms for contextual BwK and these non-contextual BwK algorithms? Are these algorithms essentially a combination of those non-contextual BwK algorithms with a learning oracle, or do them have unique characteristics of its own?
>
> Specifically, does the problem you are studying have a counterpart in the non-contextual BwK setting (i.e., BwK under the small budget scenario)? If so, what is the relationship between your algorithm and the algorithms for non-contextual BwK under the small budget scenario? Is your algorithm essentially a combination of those non-contextual BwK algorithms with a learning oracle, or does it have unique characteristics of its own?

---

> > ### Author Response · Authors · 2024-11-27
> > **Response to Reviewer Nbbq**
> >
> > We apologize for the possible misunderstanding. The short answer is *the existing algorithms are different for BwK and CBwK, but our algorithm is the "same" for BwK and CBwK*. Let's detail it below.
> >
> > CBwK is more challenging than BwK due to the unknown distribution of stochastic contexts (its unique characteristic) in the small budget regime. The existing algorithms for CBwK require this information to be taken into account. It goes beyond a combination of non-contextual BwK algorithms with learning oracles. Let's illustrate this with two classical algorithms: the primal-dual algorithm with UCB for BwK in Badanidiyuru et al. (2018) and the primal-dual algorithm with linear UCB in Agrawal & Devanur (2016). The work in Badanidiyuru et al. (2018) initiated the study of (non-contextual) BwK and proposed a primal-dual method with UCB/LCB estimators. When generalizing this approach to (linear) CBwK, Agrawal & Devanur (2016) integrated linear UCB/LCB estimators into the primal-dual template. However, Agrawal & Devanur (2016) required an additional exploration process to estimate the optimal value of the underlying offline problem, which encodes knowledge of the context distribution.
> >
> > In contrast, our design takes a different approach. While there is a non-contextual counterpart of BwK under a small budget, unlike the two examples above, *our algorithm for BwK and CBwK remains the "same" due to its single-stage, budget-aware adaptive design (our unique characteristic).* Specifically, the adaptive design implicitly learns the context distribution without requiring additional learning procedures. In other words, when applied to BwK, our algorithm naturally reduces to a budget-aware primal-dual algorithm with UCB/LCB. Finally, we want to emphasize that our design is inspired by the general primal-dual template but incorporates a novel adaptive budget-aware design and theoretical analysis.
> >
> > We hope this addresses your question, and please let us know if you have any further questions.
> >
> > ---
> >
> > ### References
> >
> > - Ashwinkumar Badanidiyuru, Robert Kleinberg, and Aleksandrs Slivkins. *Bandits with knapsacks*. *Journal of the ACM*, 2018.
> > - Shipra Agrawal and Nikhil Devanur. *Linear contextual bandits with knapsacks*. *In Advances in Neural Information Processing Systems*, 2016.

---

> > > ### Comment · Reviewer_Nbbq · 2024-11-27
> > >
> > > Thanks for your timely reply. I have no further concerns!

---

> > > > ### Author Response · Authors · 2024-11-27
> > > >
> > > > Thank you very much for your time!

---

### Meta-Review · Area_Chair_FN3R · 2024-12-22

**Metareview:**

This paper addresses the problem of stochastic contextual bandit with knapsack, focusing on scenarios where the budget $B$ is smaller than the time horizon $T$. It presents an effective algorithm and theoretical analysis tailored to such scenarios and validates their efficacy through experiments. The paper tackles a natural problem formulation that is likely to attract the interest of the community. Strengths include the proposed algorithm's independence from prior knowledge of feasibility assumptions.

However, there are concerns such as the lack of intuitive explanations for why algorithms from prior studies are not applicable in settings with small budgets and why the Lyapunov drift method is effective, as well as the presence of numerous typos.

With the premise that these concerns will be addressed in the camera-ready version, I support the acceptance of this paper.

**Additional Comments On Reviewer Discussion:**

In the reviews, concerns were raised regarding the lack of intuitive explanations for why algorithms from prior studies are not applicable in settings with large budgets and why the Lyapunov drift method is effective, as well as the presence of numerous typos. However, no further concerns were expressed in response to the authors' rebuttal.

---

### Decision · Program_Chairs · 2025-01-22

Accept (Poster)